# Molecular Mechanisms of CBL-CIPK Signaling Pathway in Plant Abiotic Stress Tolerance and Hormone Crosstalk

**DOI:** 10.3390/ijms25095043

**Published:** 2024-05-06

**Authors:** Cengiz Kaya, Ferhat Uğurlar, Ioannis-Dimosthenis S. Adamakis

**Affiliations:** 1Soil Science and Plant Nutrition Department, Agriculture Faculty, Harran University, Sanliurfa 63200, Turkey; c_kaya70@yahoo.com (C.K.); ferhatugurlar@gmail.com (F.U.); 2Section of Botany, Department of Biology, National and Kapodistrian University of Athens, 15784 Athens, Greece

**Keywords:** calcineurin B-like proteins, CBL-interacting protein kinases, plant stress resilience, hormone interplay, genetic modification

## Abstract

Abiotic stressors, including drought, salt, cold, and heat, profoundly impact plant growth and development, forcing elaborate cellular responses for adaptation and resilience. Among the crucial orchestrators of these responses is the CBL-CIPK pathway, comprising calcineurin B-like proteins (CBLs) and CBL-interacting protein kinases (CIPKs). While CIPKs act as serine/threonine protein kinases, transmitting calcium signals, CBLs function as calcium sensors, influencing the plant’s response to abiotic stress. This review explores the intricate interactions between the CBL-CIPK pathway and plant hormones such as ABA, auxin, ethylene, and jasmonic acid (JA). It highlights their role in fine-tuning stress responses for optimal survival and acclimatization. Building on previous studies that demonstrated the enhanced stress tolerance achieved by upregulating CBL and CIPK genes, we explore the regulatory mechanisms involving post-translational modifications and protein–protein interactions. Despite significant contributions from prior research, gaps persist in understanding the nuanced interplay between the CBL-CIPK system and plant hormone signaling under diverse abiotic stress conditions. In contrast to broader perspectives, our review focuses on the interaction of the pathway with crucial plant hormones and its implications for genetic engineering interventions to enhance crop stress resilience. This specialized perspective aims to contribute novel insights to advance our understanding of the potential of the CBL-CIPK pathway to mitigate crops’ abiotic stress.

## 1. Introduction

Abiotic stressors such as drought, salt, cold, and heat exert profound impacts on plant growth and development. These stressors trigger a cascade of cellular responses, and among the key players orchestrating plant adaptation and resilience is the CBL-CIPK pathway [1,2,3]. The CBL-CIPK pathway comprises two essential elements: calcineurin B-like proteins (CBLs) and CBL-interacting protein kinases (CIPKs) [4]. CBLs form a family of calcium sensors that perceive calcium fluctuations within the plant cell in response to various abiotic stressors, such as drought, salt, and temperature extremes. This family consists of multiple members, each with distinct patterns of expression and specificity for different stress conditions, allowing plants to respond adaptively to diverse challenges. CIPKs, on the other hand, are serine/threonine (Ser/Thr) protein kinases that function downstream of CBLs [4].

Upon binding to CBLs, CIPKs are activated and can transmit calcium signals to downstream targets such as transporters, ion channels, enzymes, and transcription factors. This activation modulates various cellular processes, including ion homeostasis, osmotic balance, and gene expression, which are essential for plant adaptation to stress [5]. Moreover, the CBL-CIPK system is involved in regulating plant metabolism and development, contributing to growth even under adverse conditions [6].

Furthermore, the CBL-CIPK pathway intricately interacts with plant hormones such as ABA, auxin, ethylene, and jasmonic acid (JA), effectively integrating hormonal control into stress responses [5,7]. These interactions fine-tune hormone balance and communication, enabling plants to optimize their responses for survival and acclimatization [5,6].

Previous studies have provided insights into how the CBL-CIPK pathway enhances the ability of plants to tolerate abiotic stress. For instance, studies have shown that increasing the expression of CBL and CIPK genes can enhance stress tolerance in various plant species [8,9,10]. Additionally, the regulatory mechanisms of the CBL-CIPK pathway are linked to post-translational modifications, particularly phosphorylation, and interactions between proteins [11,12].

While previous studies have contributed significantly to our understanding of the role of the CBL-CIPK pathway in promoting abiotic stress tolerance, knowledge gaps persist. Specifically, the complex interplay between the CBL-CIPK system and plant hormone signaling under various abiotic stress conditions remains a challenge.

Distinct from the broad perspectives provided by Tang et al. [4] and Ma et al. [5], our review paper takes a focused approach to study the interaction between the CBL-CIPK signaling pathway and key plant hormones, along with its implications for genetic engineering in the context of abiotic stress responses. While Tang et al. [4] underscore the membrane-associated localization and role in the membrane transport processes of CBL proteins, Ma et al. [5] explore the conserved structure and physiological functions of CBL-CIPK networks in response to multiple stimuli. In this review, we provide a specialized examination of the CBL-CIPK signaling pathway, elucidating its synergistic interactions with key plant hormones. Unlike broader explorations of the topic, we specifically highlight the role of the pathway in abiotic stress responses and its potential for genetic engineering applications.

## 2. Structural and Functional Aspects of the CBL-CIPK Pathway

The CBL-CIPK system of plants is dependent on CBLs, specialized proteins that act as calcium sensors [4]. These CBLs can interact with calcium ions and certain CIPKs because they have conserved domains including EF-hand motifs, a NAF domain, and a C-terminal CBL domain [13]. Through their calcium-binding properties, CBLs perceive and decode intracellular calcium fluctuations, initiating downstream signaling events by activating and targeting CIPKs [5,14]. For example, under drought stress conditions, CBLs detect decreased calcium levels, and upon interaction with specific CIPKs, they regulate stomatal closure and promote water conservation [13]. Similarly, Akram et al. [15] found that *GbNHX7* interacts with CBLs and CIPKs, implying its involvement in the CBL-CIPK pathway during cotton’s salt stress. Additionally, Yu et al. [16] found 8 VaCBLs and 19 VaCIPKs in cold-tolerant Amur grape. VaCIPK18, highly induced by cold stress, rescued cold sensitivity in Arabidopsis mutants. Its overexpression enhanced cold tolerance by modulating the CBF pathway and reducing reactive oxygen species. These examples highlight the diverse roles of the CBL-CIPK pathway in coordinating plant responses to various abiotic stresses.

Furthermore, CIPKs represent Ser/Thr protein kinases known for their interaction with CBLs and act as mediators of CBL-mediated calcium signaling [17]. These protein kinases play a vital role in transmitting the calcium signals perceived by CBLs to downstream targets, facilitating appropriate plant responses to abiotic stresses [18]. CIPKs typically consist of four domains: a catalytic kinase domain, a regulatory domain containing an auto inhibitory junction and NAF/FISL motif, a junction domain, and a C-terminal domain with a targeting sequence for specific subcellular compartments [3].

The interplay between CBLs and CIPKs is central to stress perception and response in the CBL-CIPK pathway. CBLs form physical complexes with CIPKs, playing a critical role in calcium-mediated signal transduction [19]. Key domains that enhance this interaction and serve as molecular switches include the NAF domain in CIPKs and the C-terminal CBL domain in CBLs [13,20]. These domains transmit calcium signals and influence downstream targets related to stress responses.

Additionally, the activation and control of the CBL-CIPK pathway are greatly influenced by the dependency and interaction between CBLs and CIPKs. Genetic studies have shown that altered stress responses and reduced stress tolerance in plants can result from disruptions or mutations in certain CBL or CIPK genes [21,22]. To fine-tune stress perception and reaction, CBL and CIPK expression levels and activities are closely controlled in response to stress [5,23]. Additionally, post-translational changes including ubiquitination and phosphorylation help to dynamically regulate the CBL-CIPK pathway [24]. So, the interaction between CBLs and CIPKs forms crucial complexes for calcium-mediated signal transduction, emphasizing their significance in stress perception and response. Genetic studies have shown that disruptions or mutations in specific CBL or CIPK genes can modify stress responses and decrease stress tolerance in plants. Hence, understanding the regulatory mechanisms of the CBL-CIPK pathway, including post-translational modifications, aids in refining stress perception and response, thereby enhancing plant resilience against environmental challenges.

## 3. The Regulation of the CBL-CIPK Pathway

The CBL-CIPK pathway regulates plant abiotic stress tolerance through post-translational modifications, protein–protein interactions, and phosphorylation-dependent and calcium-dependent mechanisms. The contents of Table 1, detailing the diverse functions and regulatory mechanisms of the CBL-CIPK pathway in plant stress responses, are discussed in the following subsections.

### 3.1. Regulation of CBL-CIPK Pathway by Phosphorylation Events

Phosphorylation events, which affect the activation or inactivation of CBLs and CIPKs within the signaling network, closely control the CBL-CIPK pathway. The mechanism of phosphoregulation within the CBL-CIPK module has been elucidated, showing that the phosphorylation of substrates by CIPKs, activated by CBLs bound to Ca^2+^, alters downstream targets, enabling plants to adapt to environmental stresses [25]. CBL proteins function as calcium sensors, interacting with CIPK kinases to mediate the phosphorylation of downstream targets and regulate various ion transport systems and nutrient responses [26].

Under low-potassium stress, Li et al. [27] demonstrated the dynamic regulation of protein abundance and phosphorylation in CBL-CIPK-channel modules, facilitating K^+^ uptake and remobilization. CIPK9/23 kinases phosphorylate CBL1/9/2/3, while ABI1/ABI2/HAB1/PP2CA phosphatases dephosphorylate *CBL2/3-CIPK9* upon K^+^ depletion. Chu et al. [28] found that the CBL-CIPK signaling network, through phosphorylation, plays a vital role in regulating nitrate transport and root growth in *Arabidopsis*, affecting the distribution of plasma membrane microdomains and nitrate transporters.

These findings emphasize the importance of post-translational modifications, including protein phosphorylation, in the complex signaling pathways governing essential processes in plant stress responses, growth, and development through the CBL-CIPK pathway.

### 3.2. The Specificity of Calcium Signal-Mediated Regulation in the CBL-CIPK Pathway

The CBL-CIPK pathway inherently serves as a calcium-sensing pathway in plant cells, where CBL proteins detect changes in calcium concentrations and signal to CIPKs to enact specific cellular responses. However, calcium signal-mediated regulation in the CBL-CIPK pathway adds a layer of specificity to the way plants respond to a diverse range of environmental stimuli [13,29]. In this specialized regulation, CBL proteins are localized in various subcellular compartments and contain EF-hand motifs that enable precise binding to calcium ions [4,30]. This binding capacity varies among different CBL proteins, allowing them to react to specific calcium signals at varying concentrations and durations [31]. This intricate regulatory mechanism enables the fine-tuning of cellular responses based on calcium signals.

Calcium signals allow for the coordination of plant responses to multiple environmental factors, including light, temperature, pathogens, hormones, and mechanical stressors, guiding the adaptive strategies of plants for survival. The adaptability and crosstalk between CBL and CIPK proteins, along with their specific subcellular positioning, play a critical role in mediating these diverse signals [32].

Dong et al. [33] emphasize the significance of Ca^2+^ signals and the CBL-CIPK network in plant physiology and nutritional balance. They explore the role of nutrient-induced Ca^2+^ elevations as initial reactions to environmental fluctuations, which are key in modulating the activity of vital transporters and channels involved in the uptake, distribution, and storage of nutrients. Additionally, Zhang et al. [34] offer insightful explanations of the relevance of CBL-CIPK and Ca^2+^ signaling in potassium (K^+^) allocation in cotton. The GhAKT2bD, AKT2 K^+^ channel, which facilitates K^+^ distribution in the xylem and phloem, is acknowledged by them as being essential. The research shows how *AtCBL4-GhCIPK1* and *AtCBL4-AtCIPK6* calcium signaling complexes intricately govern *GhAKT2bD*. These results underline the crucial role played by CBL-CIPK and Ca^2+^ signaling in controlling K^+^ distribution, which is important for optimal plant growth and development.

In conclusion, while the CBL-CIPK pathway operates as a calcium-sensing pathway, the nuances of calcium signal-mediated regulation provide specificity and adaptability in how plants perceive and respond to environmental cues. This ensures the precise modulation of the pathway, allowing plants to acclimate effectively to diverse stresses.

### 3.3. The Regulation of the CBL-CIPK Pathway by Gene Expression and Protein Abundance

Plant functioning is impacted by the control of the CBL-CIPK pathway, which also regulates protein quantity and gene expression. In response to stimuli or developmental phases, transcription factors regulate gene expression [2,35]. The response of the CBL-CIPK pathway to numerous environmental signals is impacted by these transcriptional regulations on the availability and activity of CBL and CIPK proteins.

Instead of delving into the wider regulatory mechanisms, this discussion focuses on particular instances that illustrate the influence of these mechanisms on the CBL-CIPK pathway. For instance, Thoday-Kennedy et al. [36] highlighted how post-translational modifications, such as phosphorylation and ubiquitination, finely regulate CBL and CIPK protein activity, crucial for adapting to abiotic stresses, providing insight into the complex regulatory mechanisms governing the CBL-CIPK signaling pathway. By finely regulating the activity of CBL and CIPK proteins through processes such as phosphorylation and ubiquitination, plants can dynamically adjust their responses to environmental challenges such as cold, drought, heat, salinity, and low nutrient availability. These changes may affect the lifetime and activity of CBL and CIPK proteins, which may change how these proteins function in calcium signaling and stress responses. Additionally, during stress, post-translational changes, including phosphorylation and ubiquitination, can significantly change the stability and turnover of CBL and CIPK proteins [37]. These adjustments give plants a way to swiftly modify the amounts of CBL and CIPK proteins, fine-tuning their responses to environmental stimuli.

The activity of the CBL-CIPK pathway frequently entails interactions with other signaling pathways rather than separate signaling events. The CBL-CIPK pathway and other crosstalk and feedback mechanisms of pathways were highlighted by Yu et al. [38]. Plants can integrate many signals and tailor their responses to environmental challenges thanks to these interactions, which are crucial in regulating the expression and availability of components engaged in both pathways.

Additionally, under particular stress circumstances, the quantity and stability of CBL and CIPK proteins have a major impact on the effectiveness of the CBL-CIPK signaling cascade. According to Li et al. [27], the CBL-CIPK pathway is controlled by K^+^ levels during low-potassium stress, which has an impact on the protein levels and phosphorylation status of CBL-CIPK-channel modules. This regulatory mechanism enables plants to adapt to changing potassium availability and modulate stress responses accordingly.

Interactions among proteins are fundamental to the functionality of the CBL-CIPK pathway. Xiao et al. [14] demonstrated that an elevated expression of *CuCIPK16* in *Arabidopsis* enhanced cold tolerance through alterations in protein–protein interactions within the CBL-CIPK pathway. Additionally, Huang et al. [31] identified specific CBL-CIPK genes in honeysuckle involved in salt stress responses and predicted potential interactions between LjCBL4 and LjCIPK7/9/15/16 and SOS1/NHX1 through protein–protein interaction analysis. These interactions likely participate in the coordination and modulation of salt stress signaling in the plant. Tansley et al. [39] studied calcium decoder proteins in *Marchantia polymorpha*, finding protein–protein interactions among all CBL-CIPKs, and observed increased salt sensitivity upon knocking out CIPK-B, suggesting its role in salt signaling.

In summary, the regulation of the CBL-CIPK pathway is a multifaceted process involving gene expression, protein abundance, post-translational and post-transcriptional modifications, and critical protein–protein interactions. Plants are able to actively adapt to environmental stimuli and enhance their stress tolerance and signaling processes due to these interrelated systems. The conclusions from the many studies included here offer insightful information on the intricate regulatory system that controls the CBL-CIPK cascade in plants.

**Table 1 ijms-25-05043-t001:** Summary of key findings and implications in CBL-CIPK pathway and stress response.

Aspect/Category	Key Findings	Implications	Citation
Cold tolerance and root growth	*CuCIPK16* overexpression in *Arabidopsis* enhanced cold tolerance and modified root growth via CBL-CIPK pathway interactions.	Identifies specific protein–protein interactions that contribute to plant cold tolerance, offering potential targets for crop improvement.	[14]
Phosphoregulation mechanism	Efficient decoding of Ca^2+^ signals and modulation of downstream targets through substrate phosphorylation.	Insights into how CBL-CIPK module translates Ca^2+^ signals into biological responses, impacting ion transport and nutrient regulation.	[25]
Regulation during low-potassium stress	Dynamic regulation of protein abundance and phosphorylation in CBL-CIPK-channel modules under low-potassium stress.	Understanding of coordination of CBL-CIPK pathway of K^+^ uptake and remobilization to cope with low-potassium stress.	[27]
Root growth and nitrate flux regulation	Involvement of CBL-CIPK signaling network in regulating root growth, nitrate fluxes, and salt stress responses.	Importance of CBL-CIPK interactions in modulating root growth, nitrate transport, and plant adaptation to environmental changes.	[28]
Protein–protein interactions	Protein–protein interactions within CBL-CIPK pathway are integral for regulating salt stress responses in honeysuckle.	Highlights significance of protein–protein interactions in CBL-CIPK pathway functionality and their importance in plant stress responses.	[31]
Nutrient-specific Ca^2+^ elevations	Nutrient-specific Ca^2+^ elevations as early responses to fluctuations regulating essential channels and transporters.	Reveals how Ca^2+^ signaling helps coordinate nutrient uptake and distribution, contributing to plant nutrient homeostasis.	[33]
Potassium (K^+^) allocation in cotton	CBL-CIPK and Ca^2+^ signaling play significant roles in regulating K^+^ distribution in cotton, with specific CBL-CIPK complexes (*AtCBL4-GhCIPK1* and *AtCBL4-AtCIPK6*) influencing K^+^ allocation in xylem and phloem.	Potential targets for crop improvement in nutrient management through regulation of K^+^ distribution in cotton.	[34]
Expression Patterns	Relatively over-retained duplicates tend to show asymmetric expression, thus avoiding competition. Results of gene expression analyses provide insight into differential gene expression patterns in CBL and CIPK genes.	Understanding how gene expression patterns contribute to maintaining balanced dosage for interacting CBL and CIPK proteins.	[35]
Signaling Pathway	CBL-CIPK pathway facilitates adaptable stress responses in plants via Ca^2+^ signaling.	Understanding enhances plant resilience to environmental stressors.	[36]
Genetic knockouts	Genetic knockouts using CRISPR/Cas9 confirmed presence of two CIPKs and three CBLs in *Marchantia polymorpha* and highlighted involvement of CIPK-B in salt signaling, influencing salt sensitivity.	Demonstrates utility of genetic knockouts in uncovering roles of CIPKs and CBLs, providing valuable insights into involvement of CBL-CIPK pathway in salt signaling.	[39]

## 4. Crosstalk between the CBL-CIPK Pathway and Plant Hormone Signaling

A complicated process that includes the interaction of several signaling channels is the hormone regulation of stress responses. By preserving the balance of hormones, the CBL-CIPK pathway aids in this regulation. It makes sure the right hormone signals are translated and combined to orchestrate the right stress responses. The CBL-CIPK pathway connects hormone and calcium signaling, combining calcium signals and transferring them to genes and processes that respond to hormones. The phosphorylation of hormone receptors, transcription factors, and other essential elements in hormone signaling pathways are only a few of the complex regulatory processes that are involved in this crosstalk. Table 2 illustrates the interplay within the CBL-CIPK pathway and signaling pathways relevant to plant stress. It emphasizes the role of CBL and CIPK genes in a range of stressful situations (drought, salt, cold, and pathogens) as well as possible relationships with hormone signaling (ABA, ethylene, and JA), which may affect stress tolerance and immune responses.

### 4.1. Crosstalk between CBL-CIPK and ABA Signaling Pathways in Plant Stress Responses

Numerous environmental factors threaten plant growth and survival. For instance, in the oil persimmon genome, the study of CBL-CIPK genes identified 10 *DoCBL* and 23 *DoCIPK* genes, mainly from segmental duplication. These genes are upregulated under stress conditions, with *DoCBL5* and *DoCIPK05* likely interacting with AKT1, SNF, and PP2C, hinting at crosstalk with ABA signaling and stress-responsive genes [24].

Further studies have scrutinized the CBL-CIPK pathway for its critical role in plant stress responses and crop improvement. An intricate regulatory system involving 41 CIPK and 16 CBL genes in quinoa has been detected, revealing their diverse roles in biological activities. Key among these, *CqCBL13, CqCIPK11, CqCIPK15*, and *CqCIPK37* exhibit the upregulation of cis-regulatory elements associated with ABA and drought responses [2]. The *AtCIPK6* gene in *Arabidopsis* exhibits increased expression under salt, drought, and ABA treatments, suggesting its involvement in stress response pathways. This implicates the *AtCIPK6* promoter as a potential candidate for stress-inducible transgenic engineering [40]. Further research could elucidate the molecular mechanisms by which *AtCIPK6* contributes to stress tolerance and ABA signaling, potentially involving downstream targets and regulatory interactions within the stress response pathway. The significant role of *GmCIPK29* in soybean drought resistance has been uncovered, demonstrating its interaction with *GmCBL1*, which is activated by drought and ABA. Its overexpression in Arabidopsis increases ABA sensitivity and drought tolerance, improves stomatal function, and in soybean hairy roots, it reduces water loss and MDA levels during drought [41]. Similarly, the crosstalk between CBL-CIPK and ABA signaling pathways in soybean has been explored, shedding light on the role of *GmCIPK2* in enhancing drought tolerance and facilitating ABA-mediated stomatal closure. It is suggested that the physical interaction of *GmCIPK2* with *GmCBL1* may influence downstream targets that are part of ABA signaling and drought-responsive gene expression, contributing to the plant’s increased resilience to drought stress [42].

Figure 1 illustrates the interaction between *GmCIPK29* and *GmCBL1* in soybean, suggesting their upregulation under drought stress and potential involvement in the ABA signaling pathway. It also highlights the transfer of *TaCIPK27* from wheat to *Arabidopsis* to understand its role in ABA-mediated drought stress response. Additionally, it mentions the upregulation of *CqCIPK11* and *CqCBL13* in quinoa under drought stress, indicating their association with ABA signaling pathways in this resilient crop. In *Medicago*, Du et al. [43] investigated the crosstalk between CBL-CIPK and ABA signaling pathways identifying 58 CIPK and 23 CBL genes. Certain CIPK genes, including *MtCIPK17* (*MsCIPK11*), *MtCIPK2* (*MsCIPK3*), and *MtCIPK18* (*MsCIPK12*), displayed significant upregulation under ABA treatment; this suggests potential crosstalk between ABA and CBL-CIPK pathways. However, additional investigations are necessary to clarify the exact mechanisms through which these CIPK genes interact with ABA-responsive elements in gene promoters and downstream targets, contributing to enhanced stress tolerance in *Medicago*. Promoter analysis would be particularly valuable in this regard, as it can investigate the promoters of the identified CIPK genes to identify ABA-responsive elements and other regulatory motifs.

An interaction between the CBL-CIPK pathway and ABA signaling in response to drought stress has been identified, with the drought-responsive CBL gene, *MtCBL13*, in *Medicago truncatula* showing increased expression during the early stages of drought. Interestingly, Arabidopsis plants with a prolonged overexpression of *MtCBL13* exhibited increased drought sensitivity. This was evidenced by changes in physiological parameters such as ROS accumulation, electrolyte leakage, lipid peroxidation, and water loss rate, along with a downregulation of the stress marker genes *RD22* and *DREB2A* [44]. While the studies conducted by the researchers mentioned above have provided valuable insights into the intricate interactions between CBL-CIPK and ABA pathways, there is a need for further investigation to fully understand the underlying regulatory mechanisms and specific roles of individual CBL-CIPK genes in mediating ABA responses under different stress conditions. Clarifying these research gaps will enhance our understanding of plant responses to abiotic stresses and facilitate potential applications in crop breeding. Focused investigations into gene expression profiling, protein interaction networks, and the functional characterization of individual CBL-CIPK genes will provide insights into their roles in ABA-mediated stress responses. Such targeted research will not only fill existing knowledge gaps but also contribute to the development of stress-resilient crop varieties.

### 4.2. Crosstalk between CBL-CIPK and Auxin Signaling Pathways in Plant Stress Responses

Auxin influences development and growth, and studies have revealed that the CBL-CIPK pathway modulates auxin transporters and controls auxin-mediated root responses to abiotic stresses. The research conducted by Lu et al. [45] focused on *NbCIPK25*, a salt-responsive gene in the halophyte *Nitraria billardieri*. Their findings revealed that the overexpression of *NbCIPK25* under salt stress in *Arabidopsis* seedlings led to improved salt tolerance, accompanied by enhanced root growth and an increased number of root meristematic cells. Moreover, transgenic plants showed reduced levels of ROS and malondialdehyde while exhibiting higher proline content and upregulating the expression of genes related to proline metabolism under salt stress. The overexpression of *NbCIPK25* in *Arabidopsis* seedlings under salt stress led to phenotypic alterations, including enhanced root growth and a greater number of root meristematic cells. While these changes are consistent with auxin-mediated growth responses, the study did not directly investigate *NbCIPK25*’s role in auxin signaling pathways. Therefore, any potential link between *NbCIPK25* and auxin regulation remains hypothetical and should be explored in future research to clarify the mechanisms involved. Several microRNAs involved in salt tolerance have been identified, with tae-miR156 and novel-mir59 being particularly noteworthy. These microRNAs are believed to act as regulators of auxin-response factors and CBL-CIPK pathways, respectively, potentially aiding wheat’s ability to tolerate salt stress. This discovery provides valuable insights into the interplay between the CBL-CIPK and auxin signaling pathways, shedding light on the miRNA-mediated molecular processes that govern salt tolerance in wheat [46]. Both studies provide initial suggestions of a possible interplay between the auxin and CBL-CIPK pathways in plant stress responses. While the link is currently tenuous, these findings lay the groundwork for future research to explore this potential crosstalk more thoroughly. To advance our understanding, future studies should employ molecular genetic analysis, omics techniques, and biochemical assays to dissect the auxin–CBL-CIPK interaction and its role in stress adaptation.

### 4.3. Crosstalk between CBL-CIPK and Ethylene Signaling Pathways in Plant Stress Responses

Ethylene regulates processes like seed germination and responses to environmental stimuli [47]. The CBL-CIPK pathway affects ethylene biosynthesis and gene expression. [48,49,50].

Numerous studies have been conducted recently with the goal of elucidating the complex connections between the CBL-CIPK and ethylene signaling pathways across diverse plant species and their role in the plant’s responses to stress. For instance, Xiao et al. [50] investigated the rubber tree *Hevea brasiliensis* and identified 30 *HbCIPK* genes and 12 *HbCBL* genes. The study highlights that CBL-CIPK genes in rubber trees are ethylene-responsive. Further research, particularly in *planta* studies, is necessary to determine whether these genes contribute to an SOS pathway for salt tolerance.

*LlaCIPK*, a gene crucial for *Lepidium latifolium*, exhibits differential expression in response to hormone treatments: upregulation by ethylene and downregulation by ABA and SA, highlighting its complex role in hormone signaling pathways and its potential influence on cold stress responses [48]. Additionally, *CaCIPK1* has been implicated in ethylene-mediated stress responses in pepper plants, demonstrating its role in these responses [49].

These studies indicate that the CBL-CIPK and ethylene signaling pathways are involved in plant functions, such as growth, development, and stress reactions. Innovative solutions for crop resilience, stress tolerance, and general plant performance in dynamic situations are made easier by understanding this crosstalk.

### 4.4. Crosstalk between the Jasmonic Acid and CBL-CIPK Signaling Pathways in Plant Stress Responses

Plant stress responses involve the signaling molecule jasmonic acid (JA), which affects physiological and developmental processes [51]. The intricate interaction between JA signaling and the CBL-CIPK pathway has recently come to light, showing the system’s significance in improving stress resistance. In a study on the interplay between the CBL-CIPK and JA pathways in pepper plants, Ma et al. [49] identified the *CaCIPK1* gene as a regulator of drought stress tolerance. A total of 9 *CaCBL* and 26 *CaCIPK* genes were discovered by the researchers in pepper, and their effects on JA signaling were examined. According to these findings, the CBL-CIPK and JA pathways interact with one another during pepper’s reactions to stress. Based on these findings, their following study by Ma et al. [10] offered insights into *CaCIPK3*’s contributions to JA signaling and antioxidant defense mechanisms and further defined the function of *CaCIPK3* in drought stress resistance. *CaCIPK3*’s transcriptional regulators, CaWRKY1 and CaWRKY41, as well as CaCBL2’s positive regulator, CaCBL2, were found by Ma et al. in the study.

Together, the two investigations advance our knowledge of the molecular interactions between the CBL-CIPK and JA signaling pathways in the stress responses of pepper plants. These study findings open the door for possible applications in boosting stress resistance in crops and other economically significant plants by identifying particular genes and clarifying their functions in stress tolerance and signaling.

### 4.5. Crosstalk between CBL-CIPK and Salicylic Acid Signaling Pathways in Plant Stress Responses

In plants, salicylic acid (SA) plays a key role in regulating defensive mechanisms against a variety of pathogens and abiotic stresses [52]. Notably, several research studies have explored the intricate crosstalk between SA signaling and the CBL-CIPK pathway, shedding light on their combined contributions to stress tolerance and immune responses in different plant species. Wang et al. [53] highlighted that the overexpression of *TaCIPK27* positively regulated drought tolerance in transgenic Arabidopsis plants compared to controls, as demonstrated by various physiological indices and seed germination/survival rates under both normal and drought conditions. Additionally, *TaCIPK27* transgenic plants exhibited higher endogenous ABA levels during drought stress, and they showed increased sensitivity to exogenous ABA treatment during seed germination and seedling stages. Furthermore, the expression levels of drought stress and ABA-related genes were upregulated in *TaCIPK27* transgenic plants. These results suggest that *TaCIPK27* acts as a positive regulator under drought conditions, partially through an ABA-dependent pathway.

In *Salvia miltiorrhiza*, Wang and Li [54] identified 20 *SmCIPK* genes, predominantly induced by SA and methyl jasmonate (MeJA). *SmCIPK13* showed increased expression under salt stress, and its overexpression in *Arabidopsis* reduced salt sensitivity. These findings suggest a potential connection between the CBL-CIPK and SA pathways in *Salvia miltiorrhiza*’s stress responses.

Singh et al. [55] revealed that *AdCIPK5*, a CBL-interacting protein kinase originally identified in the wild peanut species *Arachis diogoi*, positively contributes to salt and osmotic stress tolerance when overexpressed in transgenic tobacco plants. Its expression is induced by SA and other signaling molecules, suggesting a link between CBL-CIPK and SA in stress responses. Figure 2 illustrates the involvement of several genes in the plant’s response to salinity stress, particularly through SA-mediated pathways. *AdCIPK5*, originating from *Arachis diogoi*, has been successfully introduced into transgenic tobacco, suggesting its significance in salinity stress response. Furthermore, both *SmCIPK13* in *Salvia miltiorrhiza* and *CaCIPK25* in chickpea exhibit induced overexpression under SA influence, indicating their roles in coping with salinity stress.

Meena et al. [56] focused on *CIPK25*, an ortholog of *Arabidopsis CIPK25*, in chickpea (*Cicer arietinum*). They observed that *CaCIPK25* gene expression increased in response to salt, dehydration, and various hormonal treatments, exhibiting distinct tissue-specific patterns. Through an analysis of the 5′-upstream activation sequence (5′-UAS) in Arabidopsis, they identified promoter regions guiding tissue-specific expression. Additionally, they noted that treatment with SA led to a slow increase in transcript levels, reaching a peak after 12 h before decreasing, suggesting a role for *CaCIPK25* in salinity tolerance. Overall, these studies collectively emphasize the significance of the crosstalk between the CBL-CIPK and SA pathways in modulating plant stress responses, providing valuable insights into the complex molecular interplay governing plant defense and adaptation mechanisms.

**Table 2 ijms-25-05043-t002:** The crosstalk and interactions of the CBL-CIPK pathway with plant stress-related signaling pathways.

Plant Name	Stress	Gene	Hormone	Findings	Citations
*Chenopodium quinoa* (Quinoa)	ABA, Drought	*CqCIPK*, *CqCBL*	ABA	Discovered 16 CBL and 41 CIPK genes, and detected cis-regulatory elements linked to ABA and drought, showing upregulation in response to drought stress.	[2]
*Capsicum annuum* (Pepper)	Drought	*CaCIPK*	JA signaling	*CaCIPK3* positively regulates drought stress resistance in pepper, involving JA signaling and antioxidant defense mechanisms.	[10]
*Diospyros oleifera* (Oil Persimmon)	Diverse stress conditions	*DoCBL*, *DoCIPK*	ABA signaling	Identified 10 *DoCBL* and 23 *DoCIPK* genes, showing upregulation under diverse stress conditions, potentially interacting with ABA signaling.	[24]
*Arabidopsis*	Salt, Drought, ABA	*AtCIPK6*	ABA signaling	*AtCIPK6* gene showed increased expression under salt, drought, and ABA treatment, suggesting its potential as a stress-inducible promoter.	[40]
*Glycine max* (Soybean)	Drought, ABA	*GmCIPK29*	ABA signaling	*GmCIPK29* gene showed enhanced expression with drought and ABA treatment, positively influencing ABA-mediated stomatal closure and gene expression.	[41]
*Glycine max* (Soybean)	Drought	*GmCIPK2*	ABA	*GmCIPK2* contributes to drought tolerance and stomatal closure mediated by ABA through interaction with *GmCBL1*.	[42]
*Medicago sativa* (Alfalfa)	Drought, salt and ABA	*MtCIPK*	ABA signaling	Identified 23 CBL and 58 CIPK genes, with certain CIPK genes showing upregulation under ABA treatment, suggesting crosstalk between ABA and CBL-CIPK.	[43]
*Medicago truncatula* (barrel medic)	Drought	*MtCBL*	ABA signaling	*MtCBL13* showed increased expression early in drought, negatively impacting drought tolerance through ABA-dependent signaling pathway.	[44]
*Nitraria billardieri* (nitre bush)	Salt	*NbCIPK*	Auxin signaling	*NbCIPK25* overexpression improved salt tolerance, accompanied by enhanced root growth, and promoted auxin response in presence of salt stress.	[45]
*Triticum aestivum* (Wheat)	Salt	*miRNAs*	Auxin signaling	Seven miRNAs were identified as potential regulators of CBL-CIPK and auxin responsive-factor, indicating crosstalk between CBL-CIPK and auxin pathways.	[46]
*Lepidium latifolium* (pepperweed)	Cold, Ethylene	*LlaCIPK*	Ethylene signaling	*LlaCIPK* played crucial role in interplay between ethylene signaling and cold stress responses in *Lepidium*, contributing to cold tolerance.	[48]
*Hevea brasiliensis* (rubber tree)	Ethylene, Latex tapping stress	*HbCIPK*, *HbCBL*	Ethylene signaling	*HbCBLs* and *HbCIPKs* interacted in various tissues, suggesting crosstalk between CBL-CIPK and ethylene pathways with implications in growth, development, and stress responses.	[50]
Transgenic *Arabidopsis*	Drought	*TaCIPK27*	ABA	Overexpression of *TaCIPK27* enhances drought tolerance, higher endogenous ABA levels, and increased sensitivity to ABA	[53]
*Salvia miltiorrhiza* (red sage)	Salt stress	*SmCIPK*	SA signaling	*SmCIPK13* showed increased expression under salt stress; its overexpression reduced salt sensitivity, suggesting involvement in SA signaling pathway.	[54]
*Nicotiana tabacum* (Tobacco)	Salt, Osmotic stress	*AdCIPK*	SA signaling	*AdCIPK5* positively contributed to salt and osmotic stress tolerance, induced by SA and other signaling molecules.	[55]
*Cicer arietinum* (Chickpea)	Salt, dehydration, hormonal treatments	*CIPK25* (*CaCIPK25*)	Salicylic acid (SA)	*CaCIPK25* expression increased in response to salt, dehydration, and hormonal treatments. SA treatment led to increased transcript levels, suggesting role in salinity tolerance.	[56]

**Figure 1 ijms-25-05043-f001:**
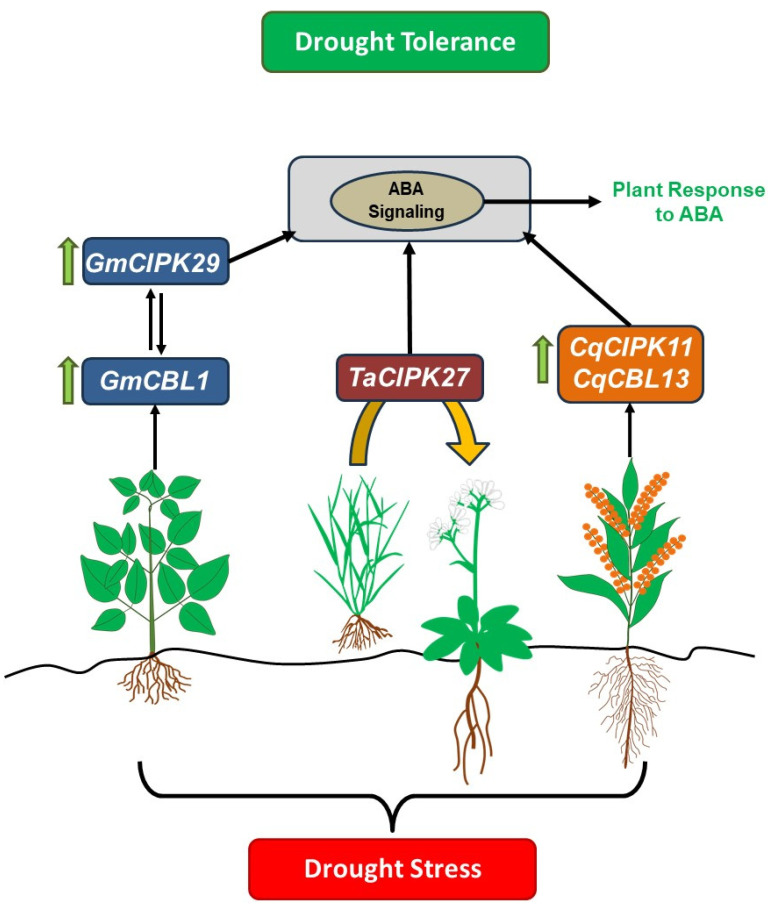
CBL-CIPK network and ABA-mediated drought response in plants. This model illustrates the interaction between *GmCIPK29-GmCBL1* in soybean, *TaCIPK27* transferred from wheat to *Arabidopsis* (indicated by the yellow arrow), and broader CBL-CIPK genes like *CqCIPK11* and *CqCBL13* in quinoa. These genes are significantly upregulated under drought stress and are associated with ABA signaling pathways [2,41,53]. Green arrows pointing upwards indicate upregulated genes.

**Figure 2 ijms-25-05043-f002:**
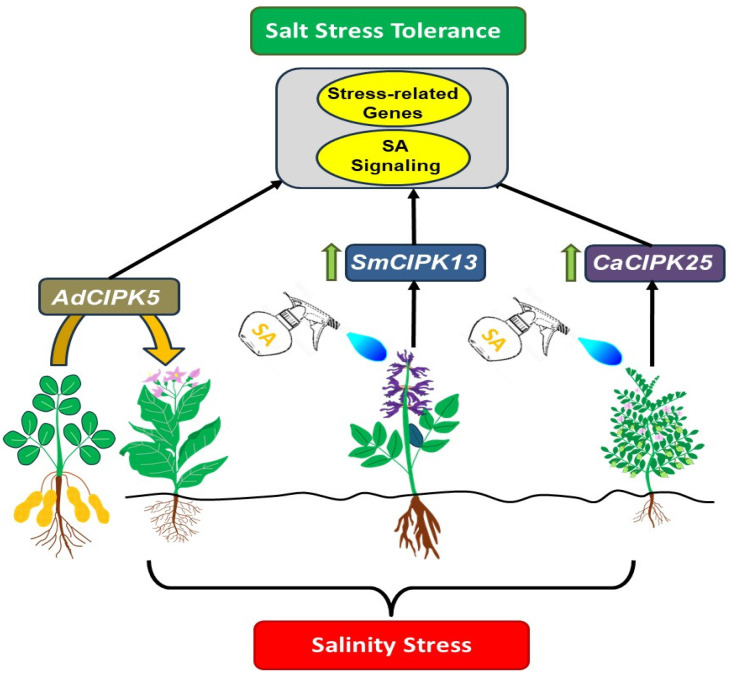
CBL-CIPK network and salicylic acid (SA)-mediated salinity stress response in plants. This model illustrates the interaction between *AdCIPK5* from the wild peanut species *Arachis diogoi*, introduced into transgenic tobacco (indicated by the yellow arrow), and the induced overexpression of *SmCIPK13* in *Salvia miltiorrhiza*, as well as *CaCIPK25* in chickpea, in response to SA. It emphasizes genes like *AdCIPK5* and *SmCIPK13*, as well as *CaCIPK25*, which are significantly upregulated under salinity stress and are linked to SA signaling pathways [54,55,56]. Green arrows pointing upwards indicate upregulated genes.

## 5. The CBL-CIPK-Mediated Transcription Factors to Modulate Stress Tolerance in Plants

The CBL-CIPK network is instrumental in orchestrating plant responses to various stress conditions through a series of downstream targets, including transcription factors (TFs) that modulate gene expression. For example, Tian et al. [57] explored the CBL-CIPK signaling network in *Kandelia obovata* seedlings. Their findings revealed 13 regulators involved in frost resistance, including transcription factors like *AP2/EREBP* and *bHLH*. These transcription factors play key roles in governing the synthesis pathways of cytokinin, ethylene, growth hormone, and flavonoids, thereby contributing to frost resistance in *K. obovata* seedlings. A possible mechanism of frost resistance in *K. obovata* may involve the modulation of ethylene and cytokinin synthesis pathways by these transcription factors, along with the regulation of flavonoid production, which collectively enhance the plant’s ability to withstand freezing temperatures.

In their investigation of cold tolerance mechanisms in *Vitis amurensis* (Amur grape), Yu et al. [16] identified the protein kinase *VaCIPK18* as a pivotal regulator within the CBL-CIPK signaling network. This study confirmed the interaction of *VaCIPK18* with all eight VaCBLs, suggesting its role as a central node in calcium signaling. *VaCIPK18* was ubiquitously expressed and highly responsive to abiotic stressors such as cold, drought, salt, and ABA. Functional assays demonstrated that *VaCIPK18* is essential for homodimerization and can rescue the cold sensitivity of *Arabidopsis Atcipk3* mutants by upregulating the CBF transcriptional pathway and reducing ROS production. Additionally, *VaCIPK18*’s interaction with *VaMYB4a* indicates a broader regulatory role in the plant’s cold response. On the subject of drought stress response, Liu et al. [58] explored the intricate network of 38 drought-responsive transcription factors (TFs) in *Seriphidium transiliense*. These TFs belonged to families such as WRKY, C2H2, AP2/ERF, MYB, bHLH, NAC, bZIP, MADS, LEA, and GRAS. The study also noted the upregulation of genes including receptor protein kinases *CRK* and *LRR-RLK*, as well as genes involved in stress signaling pathways like ABA, MAPK, and Ca^2+^ signaling, such as *CIPKs, PP2Cs*, and *CMLs*.

Turning to salt tolerance, Wang et al. [59] studied the salt tolerance mechanisms in *Prunus mume* ‘Meiren’ and unveiled a complex interplay between signaling pathways and transcription factors (TFs) that confer resilience to soil salinization. The research demonstrated that the CBL-CIPK pathway, along with mitogen-activated protein kinase, calcium-dependent protein kinase, and ABA signaling pathways, is integral to *P. mume*’s salt stress response. Key TF families such as bHLH, WRPK, ERF, and MYB were pinpointed as pivotal in modulating salt tolerance. The findings suggest that *P. mume* adapts to salt stress through a multifaceted strategy: on one hand, it employs a suite of TFs to regulate gene expression that bolsters cellular defenses and metabolic adjustments; on the other, it activates specific genes that directly counteract the physiological disruptions caused by salt, such as oxidative stress and photosynthetic inhibition. In the context of plant–microbe interactions, Yuan et al. [60] emphasized the vital role of CBL-CIPK and transcription factors (AtSRs/CAMTAs) in decoding calcium signals. Stress-triggered Ca^2+^ spikes influenced the Ca^2+^-CaM-AtSR1 complex, orchestrating plant immune responses. The study’s dynamic exploration of downstream Ca^2+^ signaling and its impact on TFs during plant defense provided valuable insights into this intricate process.

Lastly, Gratz et al. [61] investigated CBL-CIPK signaling and TFs in plant iron acquisition. The study highlights the importance of ROS, CBL-CIPK signaling, and TFs in controlling iron acquisition TF FIT. Calcium affects EHB1-mediated IRT1-mediated iron import, affecting CBL-CIPK signaling. The CBL-CIPK network is crucial for the stress tolerance of plants, influencing downstream targets and enabling adaptation and survival.

## 6. Molecular Insights into CBL-CIPK-Mediated Abiotic Stress Tolerance

Understanding abiotic stresses like salinity, drought, cold, and heat is crucial for developing stress-tolerant crop varieties and global food security. CBL-CIPK gene families regulate plant responses to these environmental conditions. In Table 3, a comprehensive overview is provided, summarizing studies that have investigated the crucial role of CBL-CIPK genes in bolstering both salinity and drought stress tolerance across diverse plant species.

### 6.1. Molecular Insights into CBL-CIPK-Mediated Salinity Stress Tolerance

CBL-CIPK genes are crucial for plant survival in salinity stress, maintaining cellular integrity and ion homeostasis. CBL-CIPK genes appear to have a role in modulating responses to salinity stress in multiple situations where they are increased in response to salinity stress. The functional genes *LjCBL2/4* and *LjCIPK1/15/17*, for instance, were identified to be increased in a study on honeysuckle by Huang et al. [31] to deal with salt stress, suggesting a potential role for these genes in the plant’s response to such conditions. The study also reveals a complex regulatory network linked to salt stress signaling through relationships between LjCBL4, LjCIPK7, LjCIPK7/9/15/16, and SOS1/NHX1. The work highlights the significance of CBL-CIPK genes in honeysuckle’s response to salt stress and recommends more investigation utilizing protein–protein interactions analyses or gene knockout studies.

Four sodium transporters involved in the response of cotton plants to salt stress were identified by Akram et al. [15]. One of these, *GbNHX7*, has interactions with both CBLs and CIPKs, indicating that it participates in the CBL-CIPK pathway. This improves salt tolerance and maintains ionic equilibrium. Further investigation, maybe using gene deletion or overexpression investigations, is required to comprehend the intricate relationships and functional importance of *GbNHX7* in cotton’s response to salt stress.

Zang et al. [62] also found links between 6 BnaCBLs and 17 BnaCIPKs in canola, highlighting their involvement in signaling pathways. Further research on functional divergence and stress tolerance contributions is needed to understand CBL-CIPK gene activities for crop development.

Similar to this, Mao et al. [63] investigated the reactions of the tobacco CBL gene family to salt and drought stress. They identified distinctive features, such as *NtCBL4A-1*’s unanticipated contribution to increased salt sensitivity, which raised concerns regarding its regulation mechanisms and the potential complexity of the CBL gene family’s role in stress responses. The specific roles and interactions of *NtCBL4A-1* and other CBL genes in tobacco, particularly in the setting of allopolyploid plant species, require more study. Understanding the exact partners and downstream targets of *NtCBL4A-1* in the CBL-CIPK pathway may shed light on the specific roles it plays in tobacco.

In poplar, Tian et al. [64] identified key *PtNHX* genes, shedding light on their roles in plant stress responses and development, including salinity. *PtNHX7*’s highlighted involvement in the CBL-CIPK pathway during salt stress responses adds to our understanding of the complex regulatory networks in poplar. Furthermore, comparing the functions and responses of *PtNHX* genes with other NHX family members in different plant species might reveal conserved or species-specific aspects of their roles in salinity tolerance.

The significance of studying CBL-CIPK gene functions in salinity stress extends beyond specific plant species. Arab et al. [65] investigated the CBL and CIPK calcium sensor families in *Aeluropus littoralis*, a halophyte plant known for its salt stress resistance. The identification of several *AlCBL* and *AlCIPK* genes and tissue-specific correlations provided valuable insights into the role of these calcium sensor families in salt stress resistance. A more comprehensive understanding of salt stress tolerance in *A. littoralis* requires deeper examinations of specific stress-responsive mechanisms and *AlCBL*-*AlCIPK* interactions. Transcriptomic or proteomic studies under varying salinity levels and time points could unveil dynamic changes in the gene expression and protein abundance of the *AlCBL* and *AlCIPK* genes in response to salt stress. Furthermore, investigating the subcellular localization of these calcium sensor proteins under stress conditions may offer insights into their spatial regulation and functional significance in cellular signaling pathways.

Furthermore, in wheat, Imtiaz et al. [66] delved into the role of the *AtCIPK16* gene in salinity stress responses. Through the generation of transgenic wheat lines expressing *AtCIPK16*, they demonstrated improved tolerance to high salinity levels. The overexpressing lines were found to retain more K^+^ ions in root tissues after salt treatment, highlighting *AtCIPK16*’s positive role in maintaining ionic homeostasis under salt stress. These findings provide valuable insights into potential strategies for sustainable wheat crop production under challenging environmental conditions. Understanding the molecular mechanisms underlying *AtCIPK16*’s positive effects on salinity tolerance could inform the development of stress-resistant wheat varieties.

Similarly, Li et al. [67] focused on eggplant, a crop limited by soil salinization in greenhouse cultivation. They investigated the role of CBLs and CIPKs and identified 5 CBL and 15 CIPK genes in eggplant. The study also revealed new CBL-CIPK complexes specific to eggplant not found in other plant species through interaction studies. Examining the expression of these genes under different ion stresses (Na^+^, K^+^, Mg^2+^, Cl^−^, and SO_4_^2−^) unveiled distinct response patterns. Notably, SmCIPK3, -24, and -25 showed higher expression levels in response to Mg^2+^ and Cl^−^ than to Na^+^ and SO_4_^2−^, respectively, shedding light on the unique ion tolerance mechanisms in eggplant. These findings offer potential for biotechnological breeding to enhance eggplant’s salinity tolerance.

Yin et al. [68] further expanded the exploration of CBL-CIPK complexes, identifying 51 *BrrCIPK* and 19 *BrrCBL* genes in turnip and grouping them into distinct clusters based on phylogenetic analysis. The proliferation of these gene families was attributed to segmental replication, leading to functional differentiation among paralogous genes over evolutionary time. They proved that *BrrCBL9.2* increases salt tolerance but BrrCBL9.1 does not through overexpression and complementation studies in *Arabidopsis*. These results shed important light on the functions of CBL-CIPK genes in turnip and highlight the significance of paralog genes in the adaptation of turnips to the harsh Qinghai–Tibet Plateau environment.

To better understand salt stress tolerance, Wu et al. [69] investigated the function of the Na^+^/H^+^ antiporter (NHX) genes in sugar beet (*Beta vulgaris* L.) under various NaCl concentrations. Five potential *BvNHX* genes were identified and categorized into the Vac-, Endo-, and PM-class NHX. Notably, *BvNHX5* was predicted to interact with CBL and CIPK, indicating that under salinity, it participates in the CBL-CIPK pathway, a signaling mechanism involved in ion transport regulation and stress responses in plants. These findings contribute to our understanding of the regulatory networks responsible for the salt stress response in sugar beet.

Overall, our understanding of CBL-CIPK interactions and gene activities in the context of salinity stress has been further enhanced by studies in canola, poplar, wheat, eggplant, turnip, and sugar beet. Studies highlight intricate regulatory networks controlling ion homeostasis and stress signaling in plants exposed to salinity.

**Table 3 ijms-25-05043-t003:** Summary of studies investigating role of CBL-CIPK gene expression and protein–protein interactions in enhancing salinity and drought stress tolerance across different plant species.

Plant Species	Genes Investigated	Key Findings	Experimental Approaches	Citation
*Carya illinoinensis* (Pecan)	30 CIPK genes, 9 CBL genes	Classification into clades based on evolutionary connections; Involvement in controlling drought response; Identification of key genes through coexpression network and qRT-PCR verification	Evolutionary analysis, coexpression network, qRT-PCR	[13]
*Gossypium hirsutum* (Cotton)	*GbNHX7*	Interaction with CBLs and CIPKs, contributing to = CBL-CIPK pathway	Protein–protein interaction analysis	[15]
*Lonicera japonica* (Honeysuckle)	*LjCBL2/4*, *LjCIPK1/15/17*	Upregulated genes essential for salt stress tolerance	Gene expression analysis	[31]
*Brassica napus* (Canola)	*BnaCBLs*, *BnaCIPKs*	Involvement in signaling pathways	Database mining, cDNA cloning, phylogenetic analysis, subcellular localization using GFP, yeast two-hybrid assay, bimolecular fluorescence complementation (BiFC), quantitative RT-PCR	[62]
*Nicotiana tabacum* (Tobacco)	*NtCBLs*	*NtCBL4A-1*’s unexpected contribution to increased salt sensitivity	Gene deletion or overexpression investigations	[63]
*Populus euphratica* (Poplar)	*PtNHXs*	*PtNHX7*’s highlighted involvement in CBL-CIPK pathway during salt stress responses	Genome identification, expression analysis, cis-acting elements assessment, coexpression network analysis, protein–protein interaction network analysis, natural variation analysis	[64]
Halophyte (*Aeluropus littoralis*)	*AlCBLs*, *AlCIPKs*	Identification of calcium sensor genes and tissue-specific correlations	Transcriptomic or proteomic studies	[65]
*Triticum aestivum* (Wheat)	*AtCIPK16*	Improved tolerance to high salinity levels	Generation of transgenic wheat lines	[66]
*Solanum melongena* (Eggplant)	CBLs, CIPKs	Identification of specific CBL-CIPK complexes and distinct ion response patterns	Interaction studies, gene expression analysis	[67]
*Brassica rapa* (Turnip)	*BrrCBLs*, *BrrCIPKs*	Functional divergence of paralog genes, BrrCBL9.2 increases salt tolerance	Overexpression and complementation studies	[68]
*Beta vulgaris* (Sugar Beet)	Five putative NHX genes (*BvNHX1*, *BvNHX2*, *BvNHX3, BvNHX4*, *BvNHX5/BvSOS1*)	Interaction with CBL and CIPK, involvement in CBL-CIPK pathway	Identification, phylogenetic analysis, exon/intron structure analysis, amiloride-binding site identification, protein–protein interaction prediction, qRT-PCR analysis	[69]
*Solanum tuberosum* (Potato)	*StCIPK18*	Crucial role in drought tolerance; Overexpression improves drought tolerance through reduced water loss, higher relative water content, proline buildup, and enhanced antioxidant enzyme activity; Knockout has opposite effect	Overexpression and knockout experiments in potatoes	[70]
*Triticum aestivum* (Wheat)	*TaCBLs*, *TaCIPKs*, *TaCIPK24*, *TaCIPK2*	TaCBLs and TaCIPKs interact in complex signaling network; Overexpression of *TaCIPK24* enhances salt tolerance; *TaCIPK2* upregulated in response to stress; Connections between TaCBL and *TaCIPK2* proteins	Transgenic Arabidopsis, transgenic tobacco, qRT-PCR	[71,72]
*Gossypium hirsutum* (Cotton)	*GhCIPK6*, *GhCBL-GhCIPK* network	*GhCIPK6* responsive to salt, drought, and ABA stimuli; Overexpression in Arabidopsis increases tolerance; Complex *GhCBL*-*GhCIPK* network with functional redundancy under abiotic stress	Overexpression studies, network analysis	[73,74]
*Sorghum bicolor* (sorghum)	*SbNHX7*	Part of CBL-CIPK pathway; Insights into functional divergence of SbNHX/NHE transporter genes under various abiotic stresses, including drought	Genome-wide identification, in silico analysis, amiloride-binding motif analysis, phylogenetic analysis, real-time gene expression analysis, protein–protein interaction network analysis	[75]

### 6.2. Molecular Insights into CBL-CIPK-Mediated Drought Stress Tolerance

Drought stress poses a threat to crop productivity and global food security. Yang et al. [70] discovered that the potato *StCIPK18* gene plays a crucial role in drought tolerance. Overexpression in potatoes improves drought tolerance through decreased water loss, higher relative water content, proline buildup, and enhanced antioxidant enzyme activity. The knockout of *StCIPK1*8 leads to the opposite effect, emphasizing the importance of controlling the potato’s response to drought.

Furthermore, Zhu et al. [2] conducted a thorough examination of the CBL and CIPK genes in quinoa, providing insight into their functions in abiotic challenges like drought. The study revealed that *CqCIPK11*, *CqCIPK15*, *CqCIPK37*, and *CqCBL13* are significantly stimulated by drought stress, highlighting their significance in quinoa’s ability to deal with water shortages. This study helps us understand the intricate regulatory network that forms these genes, which are essential for various biological functions, including quinoa’s responses to drought stress.

Regarding pecan, Zhu et al. [13] identified 30 CIPK genes and 9 CBL genes in pecan, which were phylogenetically divided into five and four clades, respectively. This classification into clades is relevant as it reflects the evolutionary relationships and potential functional diversification among the CBL and CIPK gene families. Similar structures and themes among the genes in each branch suggested functional conservation. Segmental duplication events in the genome allowed these gene families to grow. The genes’ various and distinct expression patterns under drought circumstances suggested that they were involved in controlling the drought response. The authors used a coexpression network to identify important genes involved in the process and verified their findings using qRT-PCR. They highlighted the *CiCBL* and *CiCIPK* genes, which exhibited both universal and tissue-specific roles, as confirmed through the coexpression network and qRT-PCR. Key genes such as *CiPaw.09G110800*, *CiPaw.10G083600*, and *CiPaw.12G140900* displayed similar expression patterns, while genes like *CiPaw.02G141100* and *CiPaw.16G114100* suggested specialized functions in staminate flowers and broader plant processes, respectively. The author specifically mentions the induction of three CiCIPK genes (*CiPaw.01G129000*, *CiPaw.07G161900*, and *CiPaw.13G065400*) by drought treatment and their highest expression levels after 15 days of drought stress. This suggests that these three CiCIPK genes are associated with the plant’s response to drought stress.

In addition, Sun et al. [71] studied wheat, defined the *TaCBL* and *TaCIPK* genes, and emphasized how they function in responses to abiotic stress, such as drought. According to the study, TaCBLs and TaCIPKs interact preferentially in the complex signaling network that is involved in wheat’s stress response. Enhancing salt tolerance was seen in transgenic Arabidopsis overexpressing *TaCIPK24*, suggesting potential gene candidates for more drought-related studies. Wang et al. [72] examined *TaCIPK2* in a different wheat research and discovered that it was upregulated in response to various stress treatments. Additionally, the researchers found connections between the TaCBL and *TaCIPK2* proteins. *TaCIPK2*’s function in drought stress responses was further shown by transgenic tobacco plants that overexpressed the protein.

Additionally, studies on cotton have shed light on the role of *GhCIPK6* in stress adaptation. *GhCIPK6* was found to be responsive to salt, drought, and ABA stimuli, and the overexpression of *GhCIPK6* in *Arabidopsis* increased tolerance to these stresses [73]. Sun et al. [74] identified a complex GhCBL-GhCIPK network with functional redundancy in cotton under abiotic stress, especially drought. The study’s findings furthered our knowledge of the structural features of the *GhCBL* and *GhCIPK* genes and laid the foundation for in-depth functional studies on plant stress responses.

Moreover, research on *Sorghum bicolor* by Hima Kumari et al. [75] identified *SbNHX7* as part of the CBL-CIPK pathway, offering insights into the functional divergence of SbNHX/NHE transporter genes under various abiotic stresses, including drought. These findings advance our understanding of the regulatory mechanisms that enable plants to cope with drought stress.

Overall, the collective findings from these studies highlight the pivotal roles of CBL-CIPK genes in enhancing drought stress tolerance in different plant species. The potential use of these genes to enhance drought tolerance in agricultural plants has been strongly supported by studies using transgenic techniques and functional assessments. These researchers’ molecular discoveries open the door for the creation of drought-resistant crop types, which is crucial for reducing the negative effects of water shortages on agricultural output and food security.

### 6.3. Molecular Insights into CBL-CIPK-Mediated Cold Stress Tolerance

For plants with a low threshold for cold, cold stress poses a serious threat to development and survival. Xiao et al. [14] investigated the function of the Ca^2+^-CBL-CIPK network in citrus plants in response to cold stress. Eight CBLs and seventeen CIPKs were found in two citrus species as a result of their analysis. We found that several genes displayed considerable regulation in response to cold stress. The CBL6-CIPK8 signaling network has become a common participant in the response to cold stress, which is of special interest. The study also showed that transgenic *Arabidopsis*’s greater cold tolerance was boosted by *CuCIPK16* overexpression. The functional functions of citrus CBLs and CIPKs in cold stress adaptation are now well understood due to these studies.

Yu et al. [16] investigated the function of CBL-CIPK genes in the Amur grape’s response to cold stress. In this cold-tolerant species, 8 *VaCBLs* and 19 *VaCIPKs* were discovered. The capacity of *VaCIPK18* to restore cold sensitivity in Arabidopsis mutants and its high level of induction by cold stress made it stand out among them. *VaCIPK18* overexpression improved cold tolerance via controlling the CBF transcriptional pathway and lowering the generation of reactive oxygen species. The CBL-CIPK network’s *VaCIPK18* component has become a significant part of the Amur grape’s capacity to respond to cold stress.

Wang et al. [76] elucidated the response of the CBL-CIPK gene family to low-temperature stress in Chinese cabbage. They found 47 CIPK genes and 18 CBL genes in the Chinese cabbage genome, and they also identified that tandem repeats and segmental duplication were the main forces for gene proliferation. In response to various treatments, such as Mg^2+^, K^+^, and low temperature, the expression patterns of duplicate genes changed. The functional divergence of duplicate genes during the evolution of Chinese cabbage is highlighted in their research, which also sheds light on the functions of the *BraCBL*-*BraCIPK* genes in dealing with low temperature stress.

The method through which *Kandelia obovata*, a cold-tolerant mangrove plant, resists frost was also investigated by Tian et al. [57]. They found genes that were differently expressed in roots, stems, and leaves before and after freezing by conducting a transcriptome analysis. Among them, cold resistance was linked to 13 regulators in the CBL-CIPK signaling network and MAPK cascade. The regulation of important pathways involved in frost resistance was also discovered to depend on transcription factors including AP2/EREBP and bHLH. Consequently, they contribute significantly to enhancing frost resistance in *K. obovata* seedlings. Within the established coexpression regulatory network among the 13 genes in *K. obovata*, strong correlations were detected among genes involved in the CBL-CIPK network and MAPK cascade. Notably, interactions such as *CRLK1* (*Catharanthus roseus* receptor-like kinase 1) with *MEKK1* (mitogen-activated protein kinase kinase kinase 1) and *MEKK1* with *MAPK3/6* highlight the intricate coordination between the CBL-CIPK network, MAPK cascade, and genes implicated in cold stress tolerance in K. obovata seedlings. The comprehension and preservation of mangrove ecosystems under cold stress are aided by this study’s fresh insights into the frost resistance mechanism of *K. obovata*. Figure 3 highlights *VaCIPK18*’s role in *Vitis amurensis*’ cold tolerance, interacting with VaCBL proteins and the CBF pathway for gene regulation and ROS balance. Its expression, increased by cold, and interaction with *VaMYB4a* indicate its regulatory significance. In *Kandelia obovata*, the CBL-CIPK and MAPK networks collaborate, with transcription factors controlling cold-responsive genes and the biosynthesis of key compounds, contributing to frost resistance.

**Figure 3 ijms-25-05043-f003:**
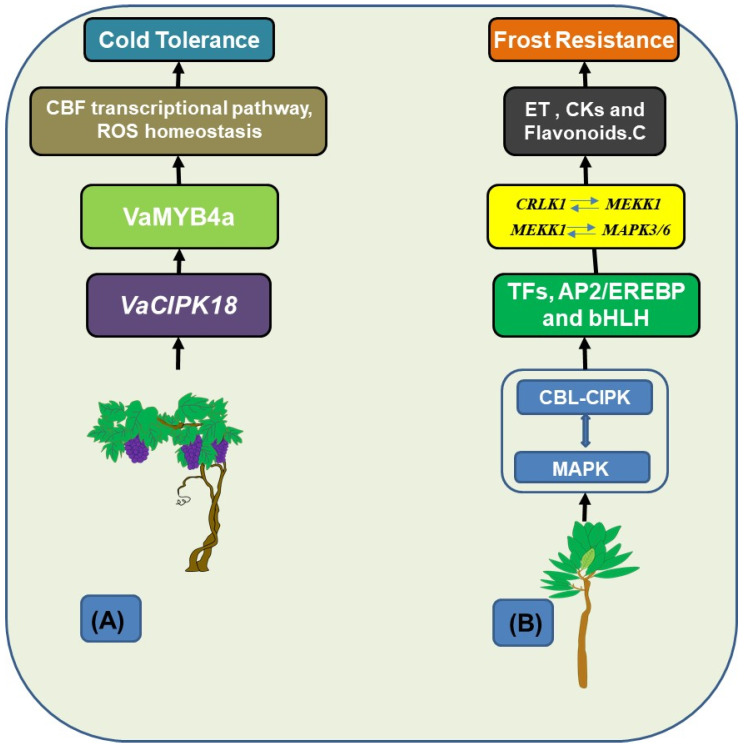
CBL-CIPK networks in the cold tolerance of *Vitis amurensis* and the frost resistance of *Kandelia obovate*. (**A**) This figure presents a comparative model illustrating the role of the CBL-CIPK signaling network in cold tolerance and frost *VaCIPK18* resistance mechanisms in two different plant species. For Vitis amurensis, it emphasizes the central role in interacting with VaCBLs to enhance cold tolerance through the C-repeat Binding Factor (CBF) transcriptional pathway and ROS homeostasis. It also notes the ubiquitous expression of *VaCIPK18* and its induction by various stresses, including cold, and its interaction with *VaMYB4a*. (**B**) It also illustrates the complex relationship between the CBL-CIPK network and the MAPK cascade in *Kandelia obovata*, highlighting their collaborative role in the regulation of transcription factors (TFs) from the AP2/EREBP and bHLH families. It details how these TFs control genes such as *CRLK1*, which interacts with *MEKK1*, and the subsequent interaction of *MEKK1* with *MAPK3/6*. These interactions are crucial for the biosynthesis of vital compounds like ethylene, cytokinin, growth hormone, and flavonoids—key elements that enhance the plant’s physiological and biochemical adaptation to cold temperatures [16,57].

Studies on citrus plants, Amur grape, Chinese cabbage, and *Kandelia obovata* have revealed the functions of CBL-CIPK genes in controlling cold-responsive pathways in response to cold stress. These results lay the groundwork for future studies to increase crops’ ability to withstand cold temperatures by offering insightful information on how various plant species adapt to low temperature circumstances.

### 6.4. Molecular Insights into CBL-CIPK-Mediated Heat Stress Tolerance

Researchers are investigating the function of the CBL-CIPK pathway in many plant species because heat stress poses a serious danger to plant development and output.

The multifaceted role of the CBL-CIPK calcium-signaling pathway in rice has been uncovered through research, revealing that *OsCBL8* has a negative impact on seed germination and seedling growth by its interaction with *OsCIPK17*. This interaction is seemingly mediated by *OsPP2C77*, a part of the ABA signaling pathway, indicating a complex regulatory mechanism at play. Further findings from the study show that *OsCBL8* and *OsCIPK17* collectively target *OsNAC77* and *OsJAMYB*, key players in rice’s defense mechanisms against high temperatures and pathogens. Such insights suggest that the CBL-CIPK pathway plays a dual role in rice, managing early developmental processes while concurrently bolstering the plant’s resistance to environmental stressors [77].

Liu et al. [78] examined the heat-stressed *CsCBL* and *CsCIPK* genes in tea plants. They discovered elements associated with stress in the promoter regions of several genes by analyzing the 7 *CsCBLs* and 18 *CsCIPKs* in the tea plant genome. *CsCBL4*, *CsCIPK2*, and *CsCIPK14* reacted to heat stress, similar to stress-related genes in *Arabidopsis*. The research further delved into the genetic structure of these genes, categorizing the CIPK family members into two groups based on intron presence. Expression profiles under various abiotic stresses were assessed using RT-qPCR, confirming the responsiveness of selected genes to such conditions. 

The mechanisms by which traditional Chinese medicinal herbs survive stress are still poorly understood. Zhang et al. [3] studied the CBL-CIPK signaling pathway in *Dendrobium catenatum*, revealing 28 CIPK genes and 9 CBL genes with tissue-specific expression patterns altered by stressors, including heat stress. Using the yeast two-hybrid system, they also discovered 10 CBL-CIPK pairings that interacted with one another.

The understanding of heat tolerance mechanisms and their molecular underpinnings is improved through the study of heat stress in rice, tea, and *Dendrobium catenatum* plants. In Table 4, a comprehensive overview is provided, summarizing studies that have investigated the crucial role of CBL-CIPK genes in bolstering both cold and heat stress tolerance across diverse plant species.

### 6.5. Molecular Insights into CBL-CIPK-Mediated Low Nutrient Tolerance

Plants are constantly exposed to fluctuating nutrient levels in their environment, which can significantly impact their growth and development [79]. To cope with these changes, plants have evolved sophisticated signaling networks that allow them to sense and respond to nutrient availability. One critical signaling pathway pivotal in plant responses to low nutrient conditions involves the CBL-CIPK network, which modulates the activity of various ion channels and transporters [7,34].

Calcium (Ca^2+^) acts as a universal second messenger in plants, mediating responses to environmental stimuli, including nutrient stress. CBL proteins are a family of Ca^2+^ sensors that decode Ca^2+^ signals elicited by nutrient deficiencies. Upon sensing elevated Ca^2+^ levels, CBLs interact with specific CIPKs to form complexes that transduce the signal downstream [80,81].

CBL-CIPK complexes are known to target ion channels and transporters, thereby regulating their activity in response to nutrient availability. For instance, under low potassium (K^+^) conditions, the MeCBL1/9-MeCIPK23 complex regulates the activity of the high-affinity potassium transporter MeAKT1, facilitating potassium uptake and signaling pathways in cassava plants [81]. Similarly, under low-K^+^ stress, Li et al. [27] elucidated the involvement of the plasma membrane CBL1/9-CIPK pathway and the tonoplast CBL2/3-CIPK pathway in promoting K^+^ uptake and remobilization, respectively, by activating a series of K^+^ channels. Furthermore, they unveiled the regulatory mechanisms linking external K^+^ levels to the activation of CBL-CIPK modules.

The role of *CBL7* in *Arabidopsis* during periods of nitrate deficiency has been elucidated by research, which shows that young seedlings’ roots increase *CBL7* expression in response to nitrate starvation. This adaptive mechanism appears to be compromised in cbl7 mutants, as evidenced by their inhibited root growth under low nitrate conditions. This inhibition is likely due to the downregulation of the high-affinity nitrate transporter genes *NRT2.4* and *NRT2.5*, suggesting that *CBL7* may act as a positive regulator of these genes, thereby facilitating nitrate uptake and assimilation [82].

**Table 4 ijms-25-05043-t004:** Summary of studies investigating role of CBL-CIPK genes in enhancing cold and heat stress tolerance across different plant species.

Plant Species	Genes Investigated	Key Findings	Experimental Approaches	Citation
*Dendrobium catenatum* (orchide)	9 CBLs, 28 CIPKs	Revealing crucial role of CBL-CIPK signaling pathway in *Dendrobium catenatum*’s abiotic stress response, including heat stress.	Study of *CBL-CIPK* signaling pathway in *Dendrobium catenatum* and its altered expression patterns under stress.	[3]
*Citrus unshiu* (satsuma mandarin) and *C. sinensis* (sweet orange)	8 CBLs, 17 CIPKs	Identification of CBL6-CIPK8 signaling network in response to cold stress. Transgenic Arabidopsis showed increased cold tolerance with *CuCIPK16* overexpression.	Investigation of Ca^2+^-CBL-CIPK network.	[14]
*Vitis amurensis* (Amur Grape)	8 *VaCBLs*, *VaCIPKs*	Discovery of VaCIPK18’s role in enhancing cold tolerance in Arabidopsis through CBF transcriptional pathway control and reduced reactive oxygen species.	Study of CBL-CIPK genes in cold-tolerant Amur grape.	[16]
*Arabidopsis*	CBLs, CIPKs	Plasma membrane CBL1/9-CIPK pathway and tonoplast CBL2/3-CIPK pathway involved in K^+^ uptake and remobilization.	Molecular insights into K^+^ response pathways and genetic analysis of phosphatase role.	[27]
*Kandelia obovata* (Mangrove)	*CBL4*, *CBL3*, *CIPK2*, *CBF1*, *CBL2*, and *MAPK6*.	Identification of genes linked to cold resistance in *K. obovata* through transcriptome analysis. Role of CBL-CIPK signaling network and MAPK cascade in frost resistance.	Investigation of frost resistance mechanisms in *K. obovata*.	[57]
*Brassica rapa* ssp. *Pekinensis* (Chinese Cabbage)	18 CBLs, 47 CIPKs	Identification of 47 CIPK genes and 18 CBL genes in response to low-temperature stress. Insights into role of *BraCBL-BraCIPK* genes in dealing with low-temperature stress.	Elucidation of CBL-CIPK gene family response to low-temperature stress in Chinese cabbage.	[76]
*Orya sativa* (Rice)	*OsCBL8*, *OsCIPK17*	Discovery of OsCBL8 interaction with OsCIPK17 to control seed germination, seedling development during heat stress, and enhance rice tolerance to extreme heat and pathogens.	Study of interaction between *OsCBL8* and *OsCIPK17* in heat stress response in rice.	[77]
*Camellia sinensis* (Tea) Plants	7 *CsCBLs*, 18 *CsCIPKs*	Identification of *CsCBL4, CsCIPK2*, and *CsCIPK1*4 as heat stress-responsive genes in tea plants. Similarities to stress-related genes in *Arabidopsis*.	Examination of heat-stressed *CsCBL* and *CsCIPK* genes in tea plants.	[78]
*Manihot esculenta* (Cassava)	*MeCBL1/9*, *MeCIPK23*, *MeAKT1*	*MeCBL1*, *MeCBL9*, *MeCIPK23*, and *MeAKT1* involved in K^+^ absorption. Differential expression of MeCBL1/9-MeCIPK23-MeAKT1 in cassava varieties under low-K^+^ stress.	In vivo role of *MeAKT1* confirmed in yeast. Differential expression analysis in cassava varieties.	[81]

## 7. Implications for Crop Stress Resilience and Future Perspectives

The CBL-CIPK pathway offers potential for enhanced crop stress resistance and long-term agricultural production through genetic engineering techniques, involving the overexpression of CBL and CIPK genes. For example, Akram et al. [15] identified sodium transporters like GbNHX7 in cotton that interact with CBLs and CIPKs, suggesting their involvement in salt stress response. In another investigation, Zang et al. [62] found interactions between 17 *BnaCIPKs* and 6 *BnaCBLs* in canola, highlighting their importance for crop improvement through multiple signaling pathways. The understanding of the CBL-CIPK pathway has led to innovative approaches for improving crop stress tolerance. It has revealed downstream targets and effectors involved in stress tolerance, such as ion transporters, channels, transcription factors, and enzymes [4,22].

Manipulating these components through genetic engineering and biotechnological strategies holds great potential for enhancing crop resilience to abiotic stresses. For instance, Yang et al. [70] investigated the *StCIPK18* gene in potato and demonstrated its crucial role in improving drought tolerance through transgenic approaches. Zhu et al. [2] shed light on the roles of CBL and CIPK genes in quinoa’s response to abiotic stresses, particularly drought, emphasizing the significance of specific *CqCIPK* and *CqCBL* genes. In citrus plants, Xiao et al. [14] explored the Ca^2+^-CBL-CIPK network’s response to cold stress, identifying key components like CBL6-CIPK8 that play important roles in cold tolerance. Furthermore, Gao et al. [77] revealed the negative regulation of seed germination and seedling growth under heat stress through the *OsCBL8*-*OsCIPK17* interaction in rice. In tea plants, Liu et al. [24] identified stress-responsive CsCBL and CsCIPK genes that can be targeted for heat stress resilience.

The potential applications of the CBL-CIPK pathway extend beyond genetic engineering approaches. Biotechnological interventions, such as genome editing technologies, offer precise and targeted modifications of the CBL-CIPK pathway components for crop improvement. CRISPR-Cas9 techniques enable the targeted manipulation of CBL or CIPK genes, enhancing their activity or modifying their regulatory properties. Tansley et al. [39] created genetic knockouts in the liverwort *Marchantia polymorpha*, confirming the existence of two CIPKs and three CBLs. The study revealed significant salt-responsive transcriptional changes in both CIPKs and one CBL, allowing for improved precision in fine-tuning stress tolerance in crops.

The ongoing investigation of this complex pathway offers promising options for sustainable agriculture and food security since it offers chances to create specialized genetic engineering plans and biotechnological solutions for boosting crop stress tolerance. Future studies should concentrate on the following areas in order to fully realize its potential:Unravel complicated relationships: Under various abiotic stress circumstances, research is required to comprehend the complex molecular interactions of the CBL-CIPK pathway and its crosstalk with other signaling pathways, such as plant hormone signaling. This knowledge will directly tailor stress response therapies.A functional characterization of the CBL and CIPK genes is crucial for stress tolerance in crop species. Gene knockout and overexpression experiments are needed to understand their roles in ion homeostasis, osmotic regulation, and gene expression.Further research is needed to identify downstream targets and effectors regulated by the CBL-CIPK pathway in various crops. This will help develop novel strategies for improved crop resilience.Future research should explore the significance of the CBL-CIPK pathway in diverse stress scenarios for crop species, enabling tailored genetic engineering strategies and biotechnological interventions for crop stress resilience.Understanding the functional divergence of CBL and CIPK genes is crucial for identifying unique stress tolerance genes and selecting candidate genes for targeted genetic modifications in different crops.Omics approaches integrate genomics, transcriptomics, proteomics, and metabolomics data to understand CBL-CIPK pathway regulation and stress responses, identifying key regulatory nodes and molecular targets for crop improvement.Validation in field conditions: Validating the findings of laboratory-based studies in field conditions is essential to assess the practical applicability of genetic engineering strategies and biotechnological interventions. Field trials will provide valuable insights into the efficacy of crop stress tolerance enhancement approaches under realistic environmental conditions.

## 8. Conclusions

In conclusion, the CBL-CIPK pathway is a central molecular orchestrator of plant abiotic stress responses. This review explored the structure, function, regulation, and crosstalk of the pathway with plant hormone signaling. It underscores the importance of the pathway in stress tolerance and the potential to enhance crop performance in challenging conditions. The calcium-dependent pathway integrates stress signals into physiological responses, influencing downstream targets such as ion channels, transporters, transcription factors, and enzymes to maintain ion homeostasis and regulate stress reactions. The CBL-CIPK pathway interacts with plant hormones like auxin, ethylene, JA, and ABA to fine-tune hormonal balance and coordinate responses to abiotic stressors. This hormone–calcium crosstalk clarifies the molecular basis of stress tolerance by identifying pathway-regulated targets, such as ion channels and transporters, which help maintain homeostasis. The pathway-controlled transcription factors alter gene expression during stress, while impacted enzymes assist in metabolic changes and detoxification. Understanding these pathways is key to developing strategies, like genetic engineering, to enhance crop stress resistance. The CBL-CIPK pathway shows great promise in improving crop performance under challenging conditions. Genetic engineering has successfully increased stress tolerance in model plants and crops, with potential for further refinement through genome editing for customized crop development.

## Data Availability

Not applicable.

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
