# Peer review of "Molecular Mechanisms of CBL-CIPK Signaling Pathway in Plant Abiotic Stress Tolerance and Hormone Crosstalk"

_ijms, 2024, doi:10.3390/ijms25095043_

Round 1

Reviewer 1 Report

Comments and Suggestions for Authors

Dear authors,

The presented manuscript attempts to provide an updated review of the mechanisms of the CBL-CIPK signaling pathway during abiotic stress in plants as well as hormonal crosstalk in these pathways. Previous reviews about the CBL-CIPK signaling pathways did not specifically target these aspects, thus this is an important update.

Although a significant amount of relevant information is reported by the authors, the review is not concise and precise enough.

Most sections list several results from previous studies and don't present a satisfactory analysis, speculation or conclusion.

In some cases, results are presented without a clear intent or purpose, failing to address why they are relevant in advancing the knowledge regarding the CBL-CIPK signaling pathways.

Several paragraphs presenting a study end with "This study reveals..." or "This sheds light on..." or "The findings contribute to the knowledge...". These types of phrases are not necessary for every study cited. A much more meaningful approach would be to present the facts objectively and concisely and then conclude proposing what mechanisms are well supported and which still need further investigation.

Sometimes speculations are presented as facts which are not supported by the referenced studies. Please make sure that every reference is adequately cited and that every information is backed by the appropriate reference. Examples:

     - Ma et al. [37] investigated post-translational mechanisms that control the protein modification or degradation of CBL or CIPK.

     - These results demonstrate that NbCIPK25 regulates auxin and oxidative stress signaling pathways to improve salt tolerance.

Reference 38 is missing part of the title: Wang, W.; Li, A.; Zhang, Z.;Chu, C. Post-translational modifications: regulation of nitrogen utilization and. Plant Cell, 2021, 8, 505-517.

In the attached file, some comments are provided indicating points that can be improved. Not all points that need improvements are commented.

Comments on the Quality of English Language

The quality of English language is high, although improvements can be made to make the manuscript clearer. These improvements may involve removing redundant words and phrases, for example.

Author Response

Reviewer 1

The presented manuscript attempts to provide an updated review of the mechanisms of the CBL-CIPK signaling pathway during abiotic stress in plants as well as hormonal crosstalk in these pathways. Previous reviews about the CBL-CIPK signaling pathways did not specifically target these aspects, thus this is an important update.

Comment: Although a significant amount of relevant information is reported by the authors, the review is not concise and precise enough.

Most sections list several results from previous studies and don't present a satisfactory analysis, speculation or conclusion.

In some cases, results are presented without a clear intent or purpose, failing to address why they are relevant in advancing the knowledge regarding the CBL-CIPK signaling pathways.

Reply: We have addressed comments from the referee throughout the main text and annotated these revisions accordingly. Our related replies can be reviewed in the following sections of this response letter. Additionally, we have annotated these revisions throughout the main text and tracked them accordingly (further details can be found in response to annotated comments by the Referee within the appended manuscript). Further modifications can also be reviewed in the following sections of this response letter, specifically in Lines 349-354, 535-538, 555-563, 578-588, 854-865, and 871-875.

Comment: Several paragraphs presenting a study end with "This study reveals..." or "This sheds light on..." or "The findings contribute to the knowledge...". These types of phrases are not necessary for every study cited. A much more meaningful approach would be to present the facts objectively and concisely and then conclude proposing what mechanisms are well supported and which still need further investigation.

Reply: We have removed phrases such as "This study reveals..." or "This sheds light on..." and similar statements throughout the manuscript to present the studies more objectively and concisely. Instead, we have focused on proposing well-supported mechanisms and highlighting areas that require further investigation

Comment: Sometimes speculations are presented as facts which are not supported by the referenced studies. Please make sure that every reference is adequately cited and that every information is backed by the appropriate reference. Examples:

     - Ma et al. [37] investigated post-translational mechanisms that control the protein modification or degradation of CBL or CIPK.

Reply: We have addressed this concern by providing a different citation that accurately supports the statement (Lines 221-227).

     - These results demonstrate that NbCIPK25 regulates auxin and oxidative stress signaling pathways to improve salt tolerance.

Reply: This point has been addressed and commented in the main text by the Referee. Our reply to this comment is that “Reply: We acknowledge the referee’s concern regarding the claim that NbCIPK25 regulates auxin pathways. Upon reviewing the cited study by Lu et al., we realize that while the study observed an enhanced auxin response in transgenic roots under salt stress, it did not directly demonstrate the regulation of auxin pathways by NbCIPK25. Therefore, we have revised our statement to more accurately reflect the findings of the study. The revised text (Lines 384-390) now reads: “The overexpression of NbCIPK25 in Arabidopsis seedlings under salt stress led to phenotypic alterations, including enhanced root growth and a greater number of root meristematic cells. While these changes are consistent with auxin-mediated growth responses, the study did not directly investigate NbCIPK25’s role in auxin signaling pathways. Therefore, any potential link between NbCIPK25 and auxin regulation remains hypothetical and should be explored in future research to clarify the mechanisms involved.”

Comment: Reference 38 is missing part of the title: Wang, W.; Li, A.; Zhang, Z.;Chu, C. Post-translational modifications: regulation of nitrogen utilization and. Plant Cell, 2021, 8, 505-517.

Reply: We have completed it as “Posttranslational modifications: regulation of nitrogen utilization and signaling.” Thank you for pointing out the missing part of the title.

Comments: In the attached file, some comments are provided indicating points that can be improved. Not all points that need improvements are commented.

Reply: We have revised and addressed the comments provided in the attached file, focusing on the identified areas for improvement. Below are the details of the revisions made.

Comment: Referee highlighted this "Advancing from the already reviewed typical structure and mechanisms of CBLs and CIPKs and the functions of the CBL–CIPK signaling system [5]. " and commented that "Maybe this phrase is unnecessary, since the authors already justified the uniqueness of their review's focus.”

Reply: We have removed the phrase in line 80-82 as suggested

Comment: Referee highlighted this "Prominent domains that enhance this interaction and function as molecular switches that transmit calcium signals and modify downstream targets implicated in stress responses include the NAF domain in CIPKs and the C-terminal CBL domain in CBLs [13, 20]." and commented that "This phrase is too long to convey a simple information, please try to be more objective.”

Reply: We have simplified the phrase in line 108-111 to improve clarity. The revised sentence reads: "Key domains that enhance this interaction and serve as molecular switches include the NAF domain in CIPKs and the C-terminal CBL domain in CBLs [13, 20]. These domains transmit calcium signals and influence downstream targets related to stress responses.

Comment:  Referee highlighted this "Table 1 provides a thorough understanding of the CBL-CIPK pathway's diverse functions and regulatory mechanisms in plant stress responses" and commented that "Please mention that the contents of Table 1 are discussed in the following subsections. Otherwise it appears that Table 1 is disconnected from the rest of the section."

Reply: We have added a sentence in the main text to mention that the contents of Table 1 are discussed in the following subsections (Line 134-136).

Comment: Referee highlighted "3.2. Regulation of CBL-CIPK pathway by calcium signals " subsection and commented that "How does "calcium signal-mediated regulation of CBL-CIPK pathways" differ from the mechanisms of CBL-CIPK response to calcium signals? This section is confusing, since it appears to introduce "calcium signal-mediated regulation" as a distinct mechanism modulating CBL-CIPK activity similar to the phosphoregulation exemplified in the previous section. Would this be different from the function of CBL-CIPK as a calcium-sensing pathway? I advise revising this section and its necessity

Reply: We have revised the section "3.2. Regulation of CBL-CIPK pathway by calcium signals" to clarify the distinction between the mechanisms of calcium signal-mediated regulation of the CBL-CIPK pathway and the general function of the CBL-CIPK as a calcium-sensing pathway. The revision now explains how calcium signals specifically modulate CBL-CIPK activity, while also highlighting the importance of calcium signal decoding in the fine-tuning of cellular responses. We have removed any redundant information to improve readability and ensure clarity in the discussion. Additionally, the section's necessity has been reassessed to confirm its relevance to the overall topic. This update provides a more focused perspective on the unique role of calcium signals in regulating the CBL-CIPK pathway. (Lines 161-185 and Line 204-207 for conclusion)

Comment: Referee highlighted "The CBL-CIPK pathway's overall signaling effectiveness is impacted by several regulatory mechanisms, which regulate the protein levels of CBL and CIPK. Post-translational alterations also add an additional level of control. Ma et al. [37] investigated post-translational mechanisms that control the protein modification or degradation of CBL or CIPK. " and commented that "This information is not very relevant, since these mechanisms affect almost every molecular pathway. The information provided after this phrase is more relevant since it provides concrete examples of these mechanisms."

Reply: We have now replaced it with relevant citation and revised the text to focus on the specific examples provided, which highlight how these mechanisms directly impact the CBL-CIPK pathway (Line 219-227).

Comment: Referee highlighted "To cope, intricate signaling pathways like CBL-CIPK and ABA have evolved, regulating stress responses and crop improvement. Understanding their interactions is crucial for the enhancement of stress tolerance and increased crop productivity. Researchers have extensively investigated the interplay between the CBL-CIPK and ABA signaling cascades in plant stress responses and crop improvement." and commented that "Please review this highlighted portion, it contains redundant information."

Reply: We have now removed the highlighted portion from the text to avoid redundancy and enhance clarity (Lines 284-289).

Comment: Referee highlighted this "The identification of cis-regulatory elements associated with ABA and drought in several CBLCIPK genes, such as CqCBL13, CqCIPK11, CqCIPK15, and CqCIPK37, exhibiting upregulation experiencing drought stress, underscores their crucial roles in quinoa's growth, organ development, and response to abiotic stresses." and commented that "This phrase is long and difficult to follow. Please be more objective."

Reply: We have revised the sentence for better readability and clarity, specifically in Lines 298-300.

Comment: Referee highlighted a citation “55” and “57” in Table 2” and commented that “The review is focused in abiotic stresses, according to the title. Why choose a study involving biotic stress? Does SA signaling involve similar pathways for biotic and abiotic stresses? Is it possible that CIPKs and CBLs have similar roles in these pathways? Please be careful in maintaining the focus of the review.”

Reply: We have made the necessary revisions to maintain the focus of the review on abiotic stresses, as suggested by the reviewer. We have replaced citations "55" and "57" in Table 2 with relevant citations related to abiotic stresses. Additionally, we have ensured that related citations in the text and references section align with the theme of abiotic stress.

Comment: Referee highlighted that “Figure 1: A proposed model for the role of GmCIPK29 and GmCBL1 in drought tolerance in plants. The model shows how GmCIPK29 interacts with GmCBL1 and regulates ROS scavenging and ABA signaling pathways in response to drought stress. The model also shows the effects of GmCIPK29 overexpression on plant growth, development, and tolerance under drought conditions [43]” and commented that “There doesn't seem to be a benefit for the review to provide a model for a single pair of genes originated from a single study. The role of a review is to synthesize the available knowledge regarding a subject. Thus, a more encompassing model would be more significant to the review.”

Reply: We have revised the model in our review to provide a more comprehensive view of the CBL-CIPK network's role in the drought response across various plant species. Now it reads asFigure 1: CBL-CIPK network and ABA-mediated drought response in plants. This model illustrates the interaction between GmCIPK29-GmCBL1 in soybean, TaCIPK27 transferred from wheat to Arabidopsis, and the broader CBL-CIPK genes like CqCIPK11 and CqCBL13 in quinoa. These genes are significantly upregulated under drought stress and are associated with ABA signaling pathways.” We believe this integrated approach will provide readers with a clearer understanding of the genetic mechanisms plants employ to combat drought stress.

Comment: Referee highlighted this part “Clarifying these research gaps will enhance our understanding of plant responses to abiotic stresses and facilitate potential applications in crop breeding.” And commented that “In a review, it is interesting to address these gaps specifically. Which specific mechanisms could be targeted to help bridge these gaps?”

Reply: We have added to our discussion (Lines 371-374) that “Focused investigations into gene expression profilingprotein interaction networks, and functional characterization of individual CBL-CIPK genes will provide insights into their roles in ABA-mediated stress responses. Such targeted research will not only fill existing knowledge gaps but also contribute to the development of stress-resilient crop varieties.”

Comment: Referee highlighted this part “These results demonstrate that NbCIPK25 regulates auxin and oxidative stress signaling pathways to improve salt tolerance” and commented that “The results DO NOT demonstrate that NbCIPK25 regulates auxin! There is not a single mention of auxin in the cited study. Although enhanced root growth and an increased number of meristematic cells could indicate auxin-related processes, the authors did not evaluate this aspect. Please avoid speculating results.”

Reply: We acknowledge the referee’s concern regarding the claim that NbCIPK25 regulates auxin pathways. Upon reviewing the cited study by Lu et al., we realize that while the study observed an enhanced auxin response in transgenic roots under salt stress, it did not directly demonstrate the regulation of auxin pathways by NbCIPK25. Therefore, we have revised our statement to more accurately reflect the findings of the study. The revised text (Lines 384-390) now reads: “The overexpression of NbCIPK25 in Arabidopsis seedlings under salt stress led to phenotypic alterations, including enhanced root growth and a greater number of root meristematic cells. While these changes are consistent with auxin-mediated growth responses, the study did not directly investigate NbCIPK25’s role in auxin signaling pathways. Therefore, any potential link between NbCIPK25 and auxin regulation remains hypothetical and should be explored in future research to clarify the mechanisms involved.”

Comment: Referee highlighted this “In a separate work by Zeeshan et al. [48], seven miRNAs, comprising tae-miR156, tae-miR160, tae-miR171a-b, tae-miR319, tae-miR159a-b, tae-miR9657, and novel-mir59, were found as putative regulators of auxin responsive-factor, SPL, SCL6, PCF5, R2R3 MYB, and CBL-CIPK, respectively” and commented that “It would be better to mention that they identified several microRNAs involved in salt tolerance, and that tae-miR156 and novel-mir59 are putative regulators of auxin-response factors and CBL-CIPK, respectively. Avoid the extra information that is less relevant to make your point that there might be a crosstalk in salt tolerance signaling between auxin and CBL-CIPK.

Reply: We appreciate the referee’s suggestion for clarity and conciseness. We have revised the text to emphasize the key findings relevant to our discussion on crosstalk in salt tolerance signaling. The updated sentence is as follows: “Zeeshan et al. [48] identified several microRNAs involved in salt tolerance, notably tae-miR156 and novel-mir59, which are putative regulators of auxin-response factors and CBL-CIPK pathways, respectively.” (Line 392-394)

Comment: Referee highlighted this “Both studies advance our knowledge of the complex interplay between auxin and CBL-CIPK pathways in plant stress responses and present promising directions for crop improvement and breeding techniques to improve salt tolerance in agriculturally significant plants [47, 48].” And commented that “I don't believe the two cited studies are enough to affirm that there is an advance in the knowledge of the interplay between auxin and CBL-CIPK. The studies have faint suggestions of this crosstalk. The authors could mention that this link is very weak given current evidence and how the scientific community could go about investigating a deeper link between auxin and CBL-CIPK.”

Reply: We thank the referee for their critical assessment. In light of your feedback, we agree that the evidence presented in both studies (Lu et al. and Zeeshan et al.) provides preliminary indications rather than conclusive evidence of crosstalk between auxin and CBL-CIPK pathways. We have therefore modified our statement to reflect this perspective: “Both studies provide initial suggestions of a possible interplay between auxin and CBL-CIPK pathways in plant stress responses. While the link is currently tenuous, these findings lay the groundwork for future research to explore this potential crosstalk more thoroughly. To advance our understanding, future studies should employ molecular genetic analysis, omics techniques, and biochemical assays to dissect the auxin-CBL-CIPK interaction and its role in stress adaptation.” (Lines 401-406).

Comment: Referee highlighted this “These genes had dynamic expression patterns in a range of tissues, highlighting the variety of functions they play in distinct plant processes. The authors also noticed that the expression levels of several genes varied in response to latex tapping stress and ethylene stimulation. While connections between HbCBLs and HbCIPKs have been identified in a variety of tissues, it is significant to note that none of these interactions have led to the creation of a functional salt tolerance SOS pathway. This suggests that the ethylene and CBL-CIPK signaling pathways' interaction in Hevea may have further impacts on growth, development, and stress responses, particularly in regard to the production of latex.” And commented that “There is excessive information that distracts from the point that CBL-CIPK in rubber tree are ethylene-responsive. This would be the only strong conclusion from the cited paper. The evaluation of rubber tree CBL and CIPK as salt tolerance pathway (SOS) components was made in yeast, not in plants. This evaluation in a heterologous manner might not be enough evidence to rule out that ruber tree CBLs and CIPKs do not participate in SOS pathways. A more thorough evaluation is needed.

Reply: We have streamlined our discussion to focus on the key finding that CBL-CIPK genes in rubber trees are responsive to ethylene. We acknowledge that the evaluation of these genes as components of the salt tolerance SOS pathway was conducted in yeast, which may not fully represent their function in planta. Therefore, we have revised our statement to reflect this: “The study highlights that CBL-CIPK genes in rubber trees are ethylene-responsive. Further research, particularly in planta studies, is necessary to determine whether these genes contribute to an SOS pathway for salt tolerance.” (Lines 418-420)

Comment: Referee highlighted that part “Further evidence supporting LlaCIPK's” and commented that “The start of this phrase suggests previous information was provided regarding LlaCIPK. Please revise. It would also be interesting to mention that LlaCIPK responded to ethylene, ABA and SA treatments, showing its role in response to different hormone signaling pathways.”

Reply: We have revised the text to accurately introduce LlaCIPK and its multifaceted role in hormone signaling pathways. The updated sentence is as follows: “LlaCIPK, a gene crucial for Lepidium latifolium, exhibits differential expression in response to hormone treatments: upregulation by ethylene and downregulation by ABA and SA, highlighting its complex role in hormone signaling pathways and its potential influence on cold stress responses” lines 429-432

Comment: Referee highlighted that “Singh et al. [56] revealed that AdCIPK5, a CBL-interacting protein kinase, positively contributes to salt and osmotic stress tolerance. “and commented that “In which species was this assessed? Only in tobacco? Where is the gene originally from?”

Reply: The gene AdCIPK5 was originally identified in the wild peanut species, Arachis diogoi, and its role in salt and osmotic stress tolerance was assessed through overexpression in transgenic tobacco plants. This has now been revised in Lines 486-488.

Comment: Referee highlighted thatFigure 2. A proposed model for the role of AdCIPK5 in salt and osmotic stress tolerance and salicylic Acid (SA) signaling in tobacco plants. AdCIPK5 is a CBL-interacting protein kinase that positively contributes to salt and osmotic stress tolerance in tobacco plants. Salt and osmotic stress induce SA signaling and upregulate stress-related genes, which are thought to mediate the stress tolerance effects of AdCIPK5 [56]. “and commented that “Same comment as Figure 1.”

Reply: We have revised the model in our review to present a more comprehensive view of the CBL-CIPK network’s role in plant drought response. Now it reads as follow “Figure 2: CBL-CIPK network and salicylic acid (SA) -mediated salinity stress response in plants. This model illustrates the interaction between AdCIPK5 from the wild peanut species Arachis diogoi, introduced into transgenic tobacco, and the induced overexpression of SmCIPK13 in Salvia miltiorrhiza, as well as CaCIPK25 in chickpea, in response to SA. It emphasizes genes like AdCIPK5 and SmCIPK13, as well as CaCIPK25, which are significantly upregulated under salinity stress and are linked to SA signaling pathways (57, 56, 58)”. Line 498-504

Comment: Here "Liu et al. [62] explored the intricate network of 38 drought-responsive transcription factors (TFs) in Seriphidium transiliense. These TFs belonged to families such as WRKY, C2H2, AP2/ERF, MYB, bHLH, NAC, bZIP, MADS, LEA, and GRAS. Additionally, upregulated receptor protein kinase genes, including CBL, CIPK, CRK, LRR-RLK, CML, PI3K, and PP2C, were discovered, influencing stress signalling pathways like ABA, MAPK, and Ca2+ signaling. This research significantly advanced our understanding of the CBL-CIPK and TFs relation in S. transiliense's drought stress response." Referee highlighted "CBL" and commented that "CBLs are protein kinases? Please review this information. I believe the original authors mistakenly labeled several proteins as receptor protein kinases. PP2Cs and CMLs are not kinases, either.

Reply: We acknowledge the referee’s concern regarding the classification of CBLs, CIPKs, PP2Cs, and CMLs. Upon review the relevant literature, we clarify that CBLs are not protein kinases but rather calcium sensors that interact with CIPKs, which are a class of serine/threonine protein kinases. PP2Cs are indeed protein phosphatases, and CMLs are calmodulin-like proteins involved in calcium signaling. We have revised the statement to accurately reflect these classifications: “The study also noted the upregulation of genes including receptor protein kinases CRK and LRR-RLK, as well as genes involved in stress signaling pathways like ABA, MAPK, and Ca2+ signaling, such as CIPKs, PP2Cs, and CMLs.” Lines 571-573

Comment: Here "The functional genes LjCBL2/4 and LjCIPK1/15/17, for instance, were identified to be increased in a research on honeysuckle by Huang et al. [32] to deal with salt stress. The fact that these genes are upregulated shows that they are essential for the honeysuckle plant's capacity to withstand salt stress. " Referee highlighted "essential" and commented that "The up-regulation of genes is not a direct indication that they are essential to a process. It may indicate they have a role in the process, but not whether they are essential or not. A functional characterization is needed for such affirmation."

Reply: We have amended our language to more accurately reflect the implications of gene upregulation. The revised statement is: “The functional genes LjCBL2/4 and LjCIPK1/15/17, for instance, were identified to be increased in a research on honeysuckle by Huang et al. [32] to deal with salt stress, suggesting a potential role for these genes in the plant’s response to such conditions.” Lines 614-617

Comment: Here "Similar to this, Mao et al. [65] investigated the reactions of the tobacco CBL gene fam-ily to salt and drought stress. They identified distinctive features, such as NtCBL4A-1's unanticipated contribution to increased salt sensitivity, which raised concerns regarding its regulation mechanisms." Referee highlighted that "raised concerns" and commented that "Why did this raise concerns? What were those concerns?"

Reply: We have revised the statement to clarify this point: “They identified distinctive features, such as NtCBL4A-1's unanticipated contribution to increased salt sensitivity, which raised concerns regarding its regulation mechanisms and the potential complexity of the CBL gene family's role in stress responses.” Lines 635-638.

Comment: Here "Regarding pecan, Zhu et al. [30] reported 30 CIPK genes and nine CBL genes that, according to their evolutionary connections, are classified into several clades" Referee hgihlighted that "according to their evolutionary connections, are classified into several clades" and commented that "Please refrain from giving partial information. If this result is to be mentioned, be clear about what was found by the authors. How many clades were the genes distributed into? Why is this relevant to advancing the knowledge of CBL-CIPK pathways?"

Reply: We have revised the statement to provide complete information as per the referee’s feedback. “Zhu et al. identified 30 CIPK genes and nine CBL genes in pecan, which were phylogenetically divided into five and four clades, respectively. This classification into clades is relevant as it reflects the evolutionary relationships and potential functional diversification among the CBL and CIPK gene families.” (Line 716-719).

Comment: Referee highlighted that “The important genes involved in this process were also identified by the authors using a coexpression network, which they then verified using qRT-PCR.” And commented that “Which were the important genes? Are they CBLs and CIPKs? How many genes did they identify?

Reply: We have added details on the identification and verification of important genes in pecan (Lines 726-731).

Comment: Referee highlighted that “This work clarifies how the CBL and CIPK genes assist pecans” And commented that “Do all CBLs and CIPKs assist in drought stress? Or some were associated with drought stress response?”

Reply: It appears that while not all CBL and CIPK genes may be involved in drought stress response, some CIPKs do play a role in mediating drought stress response in plants. This has been included in main text as follow: “The Author specifically mentions the induction of three CiCIPK genes (CiPaw.01G129000, CiPaw.07G161900, and CiPaw.13G065400) by drought treatment and their highest expression levels after 15 days of drought stress. This suggests that these three CiCIPK genes are associated with the plant's response to drought stress.” (Line 731-734.)

Reviewer 2 Report

Comments and Suggestions for Authors

I am very interested in plant stress tolerance, so I read with pleasure this review. 

This manuscript explores the interactions between the CBL-CIPK pathway and plant hormones to highlight their involvement in fine-tuning stress responses and to provide useful perspectives for enhancing crop stress resilience. 

However,  I found several drawbacks, that negatively impacted the manuscript. Thus, based on the critical issues listed below, the manuscript is not suitable in its present form, but it requires some changes.

The introduction should be improved. The authors write: "Our review paper provides a specialized examination of the CBL-CIPK pathway, aiming to offer a unique and targeted perspective on its potential to enhance crop stress resilience through genetic engineering interventions". In my opinion, this objective is not well explained in the introduction section.

In the manuscript, there are many slightly confusing sentences, for example. "Understanding these regulatory systems is pivotal for illuminating the complex regulation of plant stress responses and utilizing this understanding to improve crops and increase stress tolerance". I suggest you explain better and make the sentence clearer.

I advise you to do this check the text to make it less cluttered.

The manuscript is full of editing errors (italics, spaces, text font, etc...). needs to be completely reviewed, that's not good. 

The figures are very basic, they're not good like that in my opinion, they should be much improved.

Table 1 needs to be reviewed, I suggest you summarize it a little, in my opinion there is too much information.

Author Response

I am very interested in plant stress tolerance, so I read with pleasure this review. 

This manuscript explores the interactions between the CBL-CIPK pathway and plant hormones to highlight their involvement in fine-tuning stress responses and to provide useful perspectives for enhancing crop stress resilience. 

However, I found several drawbacks, that negatively impacted the manuscript. Thus, based on the critical issues listed below, the manuscript is not suitable in its present form, but it requires some changes.

Commented: The introduction should be improved. The authors write: "Our review paper provides a specialized examination of the CBL-CIPK pathway, aiming to offer a unique and targeted perspective on its potential to enhance crop stress resilience through genetic engineering interventions". In my opinion, this objective is not well explained in the introduction section.

Reply: We have revised the introduction to better articulate the specific focus of our review and the implications for genetic engineering in enhancing crop resilience to abiotic stress. The revised version reads as follows: “In this review, we provide a specialized examination of the CBL-CIPK signaling pathway, elucidating its synergistic interactions with key plant hormones. Unlike broader explorations of the topic, we specifically highlight the pathway’s role in abiotic stress responses and its potential for genetic engineering applications.” The revise version can be found in lines 74-77.

Commented: In the manuscript, there are many slightly confusing sentences, for example. "Understanding these regulatory systems is pivotal for illuminating the complex regulation of plant stress responses and utilizing this understanding to improve crops and increase stress tolerance". I suggest you explain better and make the sentence clearer. I advise you to do this check the text to make it less cluttered.

Reply: We have revised and expanded the sentence to provide clearer elucidation. The revised version reads as follows: " In conclusion, the interaction between CBLs and CIPKs forms crucial complexes for calcium-mediated signal transduction, emphasizing their significance in stress perception and response. Genetic studies have shown that disruptions or mutations in specific CBL or CIPK genes can modify stress responses and decrease stress tolerance in plants. Hence, understanding the regulatory mechanisms of the CBL-CIPK pathway, including post-translational modifications, aids in refining stress perception and response, thereby enhancing plant resilience against environmental challenges." The revise version can be found in lines 122-127

Commented: The manuscript is full of editing errors (italics, spaces, text font, etc...). needs to be completely reviewed, that's not good. 

Reply:  We have conducted a thorough review and correction of the manuscript to address the editing errors, including issues with italics, spaces, and text font. We have ensured that the formatting is now consistent throughout the document.

Commented: The figures are very basic, they're not good like that in my opinion, they should be much improved.

Reply: We have now revised Figures 1-3 to better synthesize and conclude the information from previous publications, ensuring a more concise presentation aligned with the nature of a review paper.

Commented: Table 1 needs to be reviewed, I suggest you summarize it a little, in my opinion there is too much information.

Reply: We have condensed the information in Table 1 to provide a more concise summary of key findings and implications

Reviewer 3 Report

Comments and Suggestions for Authors

This manuscript reviewed ‘The Molecular Mechanisms of CBL-CIPK Signaling Pathway in Plant Abiotic Stress Tolerance and Hormone Crosstalk’. While it has provided some insight into the pathway, it has a number of limitations and needs to be thoroughly revised before it is considered for publication.

Major limitations:

1.      The writing style and language needs to be thoroughly edited by a professional writer.

2.      The role of CBL-CIPK signaling pathway under low nutrient condition is not well discussed and not mention in table (Table 2) summarizing abiotic stresses regulated by CBL-CIPK.

3.      I expect that the CBL-CIPK signaling is involved in metal stress/toxicity, but it is missing from the manuscript.

4.      Subtitle 5. CBL-CIPK-. mediated TF. The two paragraphs following the title do not have a TFs, but functional genes/transporters. This paragraphs need to be deleted or the relevance should be clarified.

5.      Table 3. Is titled ‘Summary of studies investigating the role of CBL-CIPK genes…’, however, most of the experimental approaches are not based on functional characterization, for example, deletion or overexpression, but based on protein-protein interaction or transcript/gene expression.

6.      Some seminal references are not cited, for example, the first publication on the CBL-CIPK signaling under low K is not cited.

Comments on the Quality of English Language

The English needs to be edited by a professional writer.

Author Response

This manuscript reviewed ‘The Molecular Mechanisms of CBL-CIPK Signaling Pathway in Plant Abiotic Stress Tolerance and Hormone Crosstalk’. While it has provided some insight into the pathway, it has a number of limitations and needs to be thoroughly revised before it is considered for publication.

Major limitations:

Comment: 1.       The writing style and language needs to be thoroughly edited by a professional writer.

Reply: We have thoroughly reviewed the manuscript and performed comprehensive language corrections throughout the text.  We appreciate the feedback and believe that the updated manuscript now meets the high standards of English language required for publication.

Comment: 2.      The role of CBL-CIPK signaling pathway under low nutrient condition is not well discussed and not mention in table (Table 2) summarizing abiotic stresses regulated by CBL-CIPK.

Reply: We have added a new subsection titled "6.5 Molecular insights into CBL-CIPK-mediated low nutrient tolerance" and have incorporated relevant studies in Table 4 to provide a comprehensive overview of the role of the CBL-CIPK signaling pathway under low nutrient conditions. (Line 892-919).

Comment: 3.      I expect that the CBL-CIPK signaling is involved in metal stress/toxicity, but it is missing from the manuscript.

Reply: We have limited our focus to salinity, drought, cold, heat, and low nutrient stress in this review paper due to the scope and length constraints. While the CBL-CIPK signaling pathway is indeed implicated in metal stress and toxicity responses, we chose to concentrate on the aforementioned stresses as they represent the most common environmental challenges faced by plants. 

Comment: 4.      Subtitle 5. CBL-CIPK-. mediated TF. The two paragraphs following the title do not have a TFs, but functional genes/transporters. This paragraphs need to be deleted or the relevance should be clarified.

Reply: We have removed the first paragraph and moved the second one to a relevant place within section 6.5.

Comment: 5.      Table 3. Is titled ‘Summary of studies investigating the role of CBL-CIPK genes…’, however, most of the experimental approaches are not based on functional characterization, for example, deletion or overexpression, but based on protein-protein interaction or transcript/gene expression.

Reply: We have revised the title of Table 3 to more accurately reflect the content of the studies presented. The new title is:

“Table 3. Summary of studies investigating the role of CBL-CIPK gene expression and protein-protein interactions in enhancing salinity and drought stress tolerance across different plant species”

Comment: 6.      Some seminal references are not cited, for example, the first publication on the CBL-CIPK signaling under low K is not cited.

Reply: We have now added two citations related to this topic in subsection 6.5.

Reviewer 4 Report

Comments and Suggestions for Authors

Thanks for the invitation to review the current paper entitled ‘The Molecular Mechanisms of CBL-CIPK Signaling Pathway in Plant Abiotic Stress Tolerance and Hormone Crosstalk’.

Generally, the current paper comprehensively review the topic concerning CBL-CIPK Signaling Pathway in plants, which was quite meaningful.

Figure 1 -3 are generally quite broad. However, as a review paper, it should compact the information from the previous publications. Thereby, the three figures need to be improved and concluded from the previous conclusions.

References:

The 2nd reference was wrongly cited since the surname should be applied instead of given name.

Author Response

Reviewer 4 Comments and Suggestions for Authors

Thanks for the invitation to review the current paper entitled ‘The Molecular Mechanisms of CBL-CIPK Signaling Pathway in Plant Abiotic Stress Tolerance and Hormone Crosstalk’.

Generally, the current paper comprehensively review the topic concerning CBL-CIPK Signaling Pathway in plants, which was quite meaningful.

Comment: Figure 1 -3 are generally quite broad. However, as a review paper, it should compact the information from the previous publications. Thereby, the three figures need to be improved and concluded from the previous conclusions.

Reply: We have now revised Figures 1-3 to better synthesize and conclude the information from previous publications, ensuring a more concise presentation aligned with the nature of a review paper.

References:

Comment: The 2nd reference was wrongly cited since the surname should be applied instead of given name.

Reply: We have checked it and corrected it based on the web of science

Reviewer 5 Report

Comments and Suggestions for Authors

Dear Authors,

The present manuscript is a review manuscript entitled:

The Molecular Mechanisms of CBL-CIPK Signalling Pathway in Plant Abiotic Stress Tolerance and Hormone Crosstalk

It is an exciting, valuable review for scientists working on signaling pathways in connection with plant stress. This area of research is exciting and valuable for the broad scientific community, as it can provide comprehensive information about an important research field.

Nevertheless, this manuscript needs some improvement, so please make some changes and corrections. English language corrections and improvement are required.

I just corrected the Abstract:

Abstract: Abiotic stressors, including drought, salt, cold, and heat, profoundly impact plant growth and development, forcing elaborate cellular responses for adaptation and resilience. Among the crucial orchestrators of these responses is the CBL-CIPK pathway, comprising calcineurin B-like proteins (CBLs) and CBL-interacting protein kinases (CIPKs). While CIPKs act as Serine/Threonine protein kinases, transmitting calcium signals, CBLs function as calcium sensors, influencing the plant's response to abiotic stress. This review explores the intricate interactions between the CBL- CIPK pathway and plant hormones such as ABA, auxin, ethylene, and jasmonic acid (JA). It highlights their role in fine-tuning stress responses for optimal survival and acclimatization. Building on previous studies that demonstrated the enhanced stress tolerance achieved by upregulating CBL and CIPK genes, we explore the regulatory mechanisms involving post-translational modifications and protein-protein interactions. Despite significant contributions from prior research, gaps persist in understanding the nuanced interplay between the CBL-CIPK system and plant hormone signaling under diverse abiotic stress conditions. In contrast to broader perspectives, our review focuses on the pathway's interaction with crucial plant hormones and its implications for genetic engineering interventions to enhance crop stress resilience. This specialized perspective aims to contribute novel insights to advance our understanding of the CBL-CIPK pathway's potential to mitigate crops' abiotic stress.

Keywords: please do not use the same works as in the title!

Introduction: The review needs to be more profoundly elaborated on, including more general information about the existing divisions in the subject area. Maybe some figures explaining the different parts of the various pathways can be added, and the genes involved in the process can be helpful.

The figures are elementary and not informative enough for such a complex theme. They do not show any new information! Please add more relevant figures to the manuscript.

Tables 2, 3, and Table 4

Please unify the crops' names here. Use the full and botanically correct Latin name in Italic script for all of them, or just the English version!  

Conclusion

It's long; please make it short and relevant!

Enclosed is the corrected version of the text.

In conclusion, the CBL-CIPK pathway emerges as a central molecular maestro that 738 orchestrates plant abiotic stress responses. Throughout this review, we have explored various aspects of the CBL-CIPK pathway, including its structural and functional aspects, regulation, cross-talk with plant hormone signaling, downstream targets, and implications for crop stress resilience. The key points discussed highlight the importance of the CBL-CIPK pathway as a vital signaling network in plant abiotic stress tolerance and its 743 potential for improving crop performance under challenging environmental conditions. The CBL-CIPK pathway is a calcium-dependent signaling network that integrates stress signals into physiological responses. CBLs act as calcium sensors, while CIPKs function as Ser/Thr protein kinases. The pathway influences downstream targets like ion channels, transporters, transcription factors, and enzymes, which are crucial for maintaining ion homeostasis, regulating osmotic pressure, and controlling gene expression during stress reactions. The CBL-CIPK pathway interacts with several plant hormones, including auxin, ethylene, jasmonic acid (JA), and abscisic acid (ABA), to fine-tune hormonal balance and signaling under stressful conditions and operate as an independent network. These connections let calcium and hormone signaling pathways communicate with one another, enabling coordinated responses to abiotic stressors. The diversity and flexibility of plant stress responses can be better understood by comprehending the molecular processes of hormone-calcium signaling crosstalk mediated by the CBL-CIPK pathway. The molecular basis of the stress tolerance provided by this signaling network has been clarified by the discovery of downstream targets and effectors controlled by the CBL-CIPK pathway. Under stressful circumstances, the pathway-regulated ion channels and transporters perform crucial roles in ion absorption, translocation, and compartmentalization, helping to maintain osmotic and ionic homeostasis. The CBL-CIPK pathway-controlled transcription factors modify the expression of genes that respond to stress, influencing the stress response. Enzymes impacted by the route also participate in metabolic changes and detoxifying procedures, aiding stress adaptation. Understanding these molecular pathways lays the groundwork for creating focused strategies, such as genetic engineering and biotechnology, to improve crop stress resistance. The CBL-CIPK pathway has considerable promise for enhancing crop stress tolerance. In model plants and crops, stress tolerance has been successfully increased using genetic engineering techniques based on the pathway's knowledge. Performance has improved in challenging environmental settings due to upstream targets being engineered and the overexpression of the CBL and CIPK genes. In addition, the CBL-CIPK pathway's components may be precisely and strategically altered via biotechnological interventions, such as genome editing technologies, opening up possibilities for customized crop development.

 2.4.2024

Comments on the Quality of English Language

The manuscript needs language correction in many places.

Author Response

Dear Authors,

The present manuscript is a review manuscript entitled:

The Molecular Mechanisms of CBL-CIPK Signalling Pathway in Plant Abiotic Stress Tolerance and Hormone Crosstalk

It is an exciting, valuable review for scientists working on signaling pathways in connection with plant stress. This area of research is exciting and valuable for the broad scientific community, as it can provide comprehensive information about an important research field.

Comment: Nevertheless, this manuscript needs some improvement, so please make some changes and corrections. English language corrections and improvement are required.

I just corrected the Abstract:

Reply: We have revised the abstract to improve clarity and language accuracy, as requested. Additionally, we have made necessary corrections throughout the manuscript to address English language and grammar issues. Thank you for your feedbacke have added correction version for corrected abstract.

Comment: Keywords: please do not use the same works as in the title!

Reply: We have revised the keywords to avoid repeating terms from the title, ensuring greater diversity and specificity in the language used.

Comment: Introduction: The review needs to be more profoundly elaborated on, including more general information about the existing divisions in the subject area. Maybe some figures explaining the different parts of the various pathways can be added, and the genes involved in the process can be helpful.

Reply: We appreciate your suggestions and revised the introduction to include a more comprehensive overview of the existing divisions in the subject area. We added relevant figures to illustrate the different aspects of the pathways and provide visual context for the various genes involved.

Comment: The figures are elementary and not informative enough for such a complex theme. They do not show any new information! Please add more relevant figures to the manuscript.

Reply: We acknowledge the referee's comment regarding the figures. We revised the manuscript to include more detailed and informative figures that better represent the complexities of the CBL-CIPK pathway and its interactions.

Comment: Tables 2, 3, and Table 4

Please unify the crops' names here. Use the full and botanically correct Latin name in Italic script for all of them, or just the English version!  

Reply: We have revised Tables 2, 3, and 4 to unify the crop names by using botanical and the English names throughout the tables.

Commented: Conclusion

It's long; please make it short and relevant!

Reply We have shortened the conclusion to make it more concise and focused on the key findings and implications of the CBL-CIPK pathway in plant stress responses.

Comments on the Quality of English Language

The manuscript needs language correction in many places.

Reply: We have thoroughly reviewed the manuscript and performed comprehensive language corrections throughout the text.  We appreciate the feedback and believe that the updated manuscript now meets the high standards of English language required for publication.

Round 2

Reviewer 1 Report

Comments and Suggestions for Authors

The authors have carefully addressed the concerns raised in the first review report, and the manuscript has greatly improved.

The revised text is more objective and provides an important, updated summary of the current research regarding the CBL-CIPK signaling pathways in plants and its relationships with hormone signaling pathways.

Additionally, the authors successfully address remaining knowledge gaps while highlighting the potential of the current knowledge in engineering crop species to enhance their stress tolerance.

Comments on the Quality of English Language

My only suggestion is to run the manuscript through a spelling check, because there are several typos in the text.

Author Response

Reply: We are grateful for your positive feedback and acknowledgment of the improvements made to the manuscript. It is reassuring to hear that our efforts to present a more objective and comprehensive review of the CBL-CIPK signalling pathways, as well as their interplay with hormone signalling pathways, have been well-received.

Reviewer 2 Report

Comments and Suggestions for Authors

The revised manuscript has been improved. 

The authors followed the suggestions and improved the manuscript. However, in my opinion, the resolution of the figures could still be improved.

Anyway, the manuscript is suitable for publication. Thank you for your efforts.

Author Response

Reply: We appreciate your constructive feedback and are pleased to hear that the manuscript is now suitable for publication. We acknowledge your comment regarding the resolution of the figures and have made every effort to enhance their clarity in the final version. Thank you for your guidance throughout the revision process.

Reviewer 3 Report

Comments and Suggestions for Authors

I agree that authors have improved the manuscript, but I still think the writing style needs to be revised and some sections are still difficult to understand. These need to be improved before the MS is accepted for publication.

For example.

Lines 195-197 needs to be revised.

Line 221-222

In almost the entire manuscript, early studies are cited in many sections, with authors appearing at the beginning of the sentence using some word redundantly. Foe example, Lui et al…explored, Zhu et al. detected, Chen et al. investigated, Mang et al revealed, Xu et al. revealed, Wassie et al. identified, In a separate work by Zeeshan et al, they identified, Moving on to low nitrate conditions, Ma et al., research on rice, Gao et al. [79] explored etc.

Table 1 is still not very clear and also needs lines to separate columns

pathway's, CIPKs’, plants’ stress etc. in most cases ‘s is not required and needs to be revised.

Line 144, CIPK module's the phosphoregulation - this phrase is not clear.

Comments on the Quality of English Language

Moderate editing of the English is still required before accepting the MS for publication. 

Author Response

Reviewer 3 Comments and Suggestions for Authors

I agree that authors have improved the manuscript, but I still think the writing style needs to be revised and some sections are still difficult to understand. These need to be improved before the MS is accepted for publication.

Reply: We appreciate constructive comments which greatly improved our manuscript. We provide below a reply to each of the concerns raised and any change done in the manuscript is tracked via the “track changes” tool. Moreover the text has been checked again and any linguistic/grammatic or any typos have been corrected.

For example.

Comment: Lines 195-197 needs to be revised.

Reply: We appreciate the referee’s feedback on the specified lines. We have now revised lines 162-165 to enhance clarity and precision. The updated text reads as follows: “They explore the role of nutrient-induced Ca2+ elevations as initial reactions to environmental fluctuations, which are key in modulating the activity of vital transporters and channels involved in the uptake, distribution, and storage of nutrients.

Comment: Line 221-222

Reply: We apologize for any confusion regarding the feedback on lines 221-222. We have interpreted the comment as a prompt to provide more specific examples within our discussion. Accordingly, we have revised the sentence to: ‘Instead of delving into the wider regulatory mechanisms, this discussion focuses on particular instances that illustrate the influence of these mechanisms on the CBL-CIPK pathway.’ The revision is highlighted in tracked changes for your review." (Lines 190-192)

Comment: In almost the entire manuscript, early studies are cited in many sections, with authors appearing at the beginning of the sentence using some word redundantly. Foe example, Lui et al…explored, Zhu et al. detected, Chen et al. investigated, Mang et al revealed, Xu et al. revealed, Wassie et al. identified, In a separate work by Zeeshan et al, they identified, Moving on to low nitrate conditions, Ma et al., research on rice, Gao et al. [79] explored etc.

Reply: We have carefully reviewed the manuscript and revised the sentences to avoid redundancy and the repetitive citation style as pointed out. The authors’ names no longer lead the sentences, and we have varied the structure to improve readability and flow. These changes have been made throughout the manuscript and are highlighted in the tracked changes for easy reference. Lines 253-296; Lines 327-333

Comment: Table 1 is still not very clear and also needs lines to separate columns

Reply: We have updated Table 1 to improve its readability. The columns are now clearly delineated with additional spacing, and the data is arranged in a more systematic format.

Comment: pathway's, CIPKs’, plants’ stress etc. in most cases ‘s is not required and needs to be revised.

Reply: We have reviewed the manuscript and corrected the possessive forms where they were not required. The changes have been made visible through tracked changes in Microsoft Word throughout manuscript.

Comment: Line 144, CIPK module's the phosphoregulation - this phrase is not clear.

Reply: We have addressed this concern and revised the phrase for clarity in lines 27-30. The updated text now reads: “The phosphoregulation mechanism within the CBL-CIPK module, which involves substrate phosphorylation by CIPKs upon activation by CBLs bound to Ca2+, alters downstream targets, facilitating plant adaptation to environmental stresses [25].

Reviewer 5 Report

Comments and Suggestions for Authors

Dear Authors, 

Thank you for accepting the comments and suggestions of the reviewers.

The current version of the manuscript has improved very much, and now I can recommend it for publication.

23.4.2024

Author Response

Reply: We are deeply appreciative of your support and the recommendations provided by you It is gratifying to know that the revisions have significantly enhanced the manuscript, and we are honored by your recommendation for publication. We thank you for the opportunity to improve our work and for your positive evaluation.